# Adaptive Oracle-Efficient Online Learning

**Guanghui Wang**[†]**, Zihao Hu**[†]**, Vidya Muthukumar**[‡,⋆]**, Jacob Abernethy**[†]
College of Computing[†]
School of Electrical and Computer Engineering[‡]
School of Industrial and Systems Engineering[⋆]
Georgia Institute of Technology
Atlanta, GA 30339
{gwang369,zihaohu,vmuthukumar8,prof}@gatech.edu

## Abstract

The classical algorithms for online learning and decision-making have the benefit of achieving the optimal performance guarantees, but suffer from computational complexity limitations when implemented at scale. More recent sophisticated techniques, which we refer to as *oracle-efficient* methods, address this problem by dispatching to an *offline optimization oracle* that can search through an exponentially-large (or even infinite) space of decisions and select that which performed the best on any dataset. But despite the benefits of computational feasibility, oracle-efficient algorithms exhibit one major limitation: while performing well in worst-case settings, they do not adapt well to friendly environments. In this paper we consider two such friendly scenarios, (a) "small-loss" problems and (b) IID data. We provide a new framework for designing follow-the-perturbed-leader algorithms that are oracle-efficient and adapt well to the small-loss environment, under a particular condition which we call *approximability* (which is spiritually related to sufficient conditions provided in (Dudík et al., 2020)). We identify a series of real-world settings, including online auctions and transductive online classification, for which approximability holds. We also extend the algorithm to an IID data setting and establish a "best-of-both-worlds" bound in the oracle-efficient setting.

## 1 Introduction

Online learning is a fundamental paradigm for modeling sequential decision making problems (Cesa-Bianchi & Lugosi, 2006; Shalev-Shwartz, 2011; Hazan, 2016). Online learning is usually formulated as a zero-sum game between a learner and an adversary. In each round $t = 1, \ldots, T$, the learner first picks an action $x_t$ from a (finite) set $\mathcal{X} = \{x^{(1)}, \ldots, x^{(K)}\}$ with cardinality equal to $K$. In the meantime, an adversary reveals its action $y_t \in \mathcal{Y}$. As a consequence, the learner observes $y_t$, and suffers a loss $f(x_t, y_t)$, where $f : \mathcal{X} \times \mathcal{Y} \mapsto [0, 1]$. The goal is to minimize the *regret*, which is defined as the difference between the cumulative loss of the learner $\sum_{t=1}^{T} f(x_t, y_t)$, and the cumulative loss of the best action in hindsight $L_T^* = \min_{x \in \mathcal{X}} \sum_{t=1}^{T} f(x, y_t)$.

A wide variety of algorithms have been proposed for the goal of minimizing worst-case regret (without any consideration of computational complexity per iteration); see (Cesa-Bianchi & Lugosi, 2006; Shalev-Shwartz, 2011; Hazan, 2016) for representative surveys of this literature. These algorithms all obtain a worst-case regret bound of the order $O(\sqrt{T \log K})$, which is known to be minimax-optimal (Cesa-Bianchi & Lugosi, 2006). Over the last two decades, sophisticated adaptive algorithms have been designed that additionally enjoy *problem-dependent* performance guarantees, which can automatically lead to better results in friendly environments. One of the most important example for this kind of guarantees is the so-called "small-loss" bound (Hutter & Poland, 2005;

Cesa-Bianchi & Lugosi, 2006; Van Erven et al., 2014). Such a bound depends on the best cumulative loss in hindsight (i.e. $L_T^*$) instead of the total number of rounds ($T$). Thus, this bound is much tighter than the worst-case bound, especially when the best decision performs well in the sense of incurring a very small loss. Another example is the "best-of-both-worlds" bound (Van Erven et al., 2014), which results in an even tighter regret bound for independent and identically distributed (IID) loss functions.

However, all of these algorithms applied out-of-the-box suffer a *linear* dependence on the number of decisions $K$. This is prohibitively expensive, especially in problems such as network routing (Awerbuch & Kleinberg, 2008) and combinatorial market design (Cesa-Bianchi et al., 2014), where the cardinality of the decision set grows exponentially with the natural expression of the problem. Several efficient algorithms do exist, even for uncountably infinite decision sets, when the loss functions have certain special structure (such as linearity (Kalai & Vempala, 2005) or convexity (Zinkevich, 2003)). However, such structure is often absent in the above applications of interests.

Notice that the efficiency of the above specialized methods is usually made possible by assuming that the corresponding *offline* optimization problem (i.e., minimizing the (averaged) loss) can be solved efficiently. This observation motivates the *oracle-efficient online learning* problem (Hazan & Koren, 2016). In this setting, the learner has access to a black-box offline oracle, which, given a real-weighted dataset $\mathcal{S} = \{(w^{(j)}, y^{(j)})\}_{j=1}^n$, can efficiently return the solution to the following problem:

$$\operatorname*{argmin}_{x \in \mathcal{X}} \sum_{j=1}^n w^{(j)} f(x, y^{(j)}).\tag{1}$$

The goal is to design *oracle-efficient* algorithms which can query the offline-oracle $O(1)$ times each round. Concrete examples of such an oracle include algorithms for empirical risk minimization (Bishop, 2007), data-driven market design (Nisan & Ronen, 2007), and dynamic programming (Bertsekas, 2019).

As pointed out by Hazan & Koren (2016), the design of oracle-efficient algorithms is extremely challenging and such an algorithm does not exist in the worst case. Nevertheless, recent work (Daskalakis & Syrgkanis, 2016; Syrgkanis et al., 2016; Dudík et al., 2020) has introduced a series of algorithms which are oracle-efficient when certain sufficient conditions are met. Among them, the state-of-the-art method is the *generalized-follow-the-perturbed-leader* algorithm (GFTPL, Dudík et al., 2020), which is a variant of the classical follow-the-perturbed-leader (FTPL) algorithm (Kalai & Vempala, 2005). Similar to FTPL, GFTPL perturbs the cumulative loss of each decision by adding a random variable, and chooses the decision with the smallest perturbed loss as $x_t$. However, the vanilla FTPL perturbs each decision independently, which requires to generate $K$ independent random variables in total. Moreover, the oracle in (1) can not be applied here since as it cannot handle the perturbation term. To address these limitations, GFTPL only generates a noise vector of low dimension (in particular, much smaller dimension than the size of the decision set) in the beginning, and constructs $K$ *dependent* perturbations based on the multiplication between the noise vector and a *perturbation translation matrix* (PTM). Therefore, the PTM critically ensures that the computational complexity for the noise generation itself is largely reduced. Furthermore, oracle-efficiency can be achieved by setting the elements in the PTM as carefully designed synthetic losses. Dudík et al. (2020) show that a worst-case optimal regret bound can be obtained when the PTM is *admissible*, i.e., every two rows are substantially distinct. This serves as a *sufficient condition* for achieving oracle-efficiency.

While these results form a solid foundation for general *worst-case* oracle-efficient online learning, it remains unclear whether *problem-dependent*, or *data-adaptive* bounds are achievable in conjunction with oracle-efficiency. In other words, the design of a generally applicable oracle-efficient *and* adaptive online learning algorithm has remained open. In this paper, we provide an affirmative answer to this problem, and make the following contributions.

- We propose a variant of the GFTPL algorithm (Dudík et al., 2020), and derive a new sufficient condition for ensuring oracle-efficiency while achieving the *small-loss* bound. Our key observation is that while the admissibility condition of the PTM in GFTPL successfully stabilizes the algorithm (by ensuring that $\mathbb{P}[x_t \neq x_{t+1}]$ is small), it does not always enable adaptation. We address this challenge via a new condition for PTM, called approximability. This condition ensures a stronger stability measure, i.e., the ratio of $\mathbb{P}[x_t = x^{(i)}]$

and $\mathbb{P}[x_{t+1} = x^{(i)}]$ is upper-bounded by a universal constant for any $i \in [K]$, which is critical for proving the small-loss bound. In summary, we obtain the small-loss bound by equipping GFTPL with an approximable PTM, a data-dependent step-size and Laplace distribution for the perturbation noise. As a result of these changes, our analysis path differs significantly from that of Dudík et al. (2020). Our new condition of *approximability* is simple and interpretable, and can be easily verified for an arbitrary PTM. It shares both similarities and differences from the *admissibility* condition proposed in Dudík et al. (2020). We demonstrate this through several examples where one of the sufficient conditions holds, but not the other.

- We identify a series of real-world applications for which we can construct approximable PTMs: (a) a series of online auctions problems (Dudík et al., 2020); (b) problems with a small adversary action space $|\mathcal{Y}|$ (Daskalakis & Syrgkanis, 2016); and (c) transductive online classification (Syrgkanis et al., 2016; Dudík et al., 2020). This is the first-time that the small-loss bound is obtained in all of these applications. To achieve this, we introduce novel PTMs and analysis for showing the approximability condition on these PTMs.

- We achieve the "best-of-both-worlds" bound, which enjoys even tighter results when the data is IID or the number of leader changes is small. The main idea is to combine our proposed algorithm with vanilla FTL leveraging ideas from a meta-algorithm called FlipFlop introduced in Van Erven et al. (2014).

## 2 Related Work

Our work contributes to two bodies of work: oracle-efficient online learning and adaptive online learning. In this section, we briefly review the related work in these areas.

### 2.1 Oracle-efficient online learning

For oracle-efficient online learning, the pioneering work of Hazan & Koren (2016) points out that oracle-efficient methods do not exist when dealing with general hostile adversaries, which implies that additional assumptions on the problem structure have to be made. Daskalakis & Syrgkanis (2016) consider the setting in which the cardinality of the adversary's action set $\mathcal{Y}$ is finite and small, and propose to add a series of "fake" losses to the learning history based on random samples from $\mathcal{Y}$. They prove that for this setting an $O(|\mathcal{Y}|\sqrt{T})$ regret bound can be obtained. Syrgkanis et al. (2016) study the contextual combinatorial online learning problem, where each action is associated with a binary vector. They make the assumption that the loss function set contains all linear functions as a sub-class. The approach in Syrgkanis et al. (2016) constructs a set of synthetic losses for perturbation based on randomly-selected contexts, and achieves worst-case optimal bounds when all the contextual information can be obtained beforehand, or when there exists a small set of contexts that can tell each decision apart. Dudík et al. (2020) is the first work to focus on the general non-contextual setting, and propose the generalized FTPL algorithm. This algorithm generates a small number of random variables at the beginning, and then perturbs the learning history via the innter product between the PTM matrix and the random variables. The algorithm can be implemented efficiently by setting the entries of the PTM as carefully designed loss values. Niazadeh et al. (2021) consider a more complicated combinatorial setting where the offline problem is NP-hard, but a robust *approximation* oracle exists. For this case, they propose an online algorithm based on a multiplicative approximation oracle, and prove that it has low approximate regret, which is a measure weaker than regret, since it only compares with a fraction of the cumulative loss of the best decision in hindsight. Note that none of the aforementioned methods can be easily shown to adapt to friendly structure in data. Recently, several concurrent works (Block et al., 2022; Haghtalab et al., 2022a) investigate how to obtain tighter bounds oracle-efficiently in the *smoothed-analysis* setting where the distribution of data is close to the uniform distribution (Rakhlin et al., 2011; Haghtalab et al., 2022b). The main focus is to adapt to the VC dimension of the hypothesis class, rather than improve the dependence on the number of rounds $T$.

In this paper, we mainly focus on the so-called the learning with expert advice setting (Cesa-Bianchi & Lugosi, 2006), where the action set is discrete, and the loss can be highly non-convex. On the other hand, efficient algorithms can be obtained even for continuous action sets when the loss functions have certain properties, such as linearity (Kalai & Vempala, 2005; Hutter & Poland, 2005; Awerbuch

& Kleinberg, 2008), convexity (Zinkevich, 2003; Hazan et al., 2007) or submodularity (Hazan & Kale, 2012). Finally, we note that, in this paper we mainly focus on the full-information setting, where the learner can observe the whole loss function after the action is submitted. Oracle-efficient online learning has also been widely studied in the contextual bandit setting (Langford & Zhang, 2008; Dudik et al., 2011; Agarwal et al., 2014; Foster et al., 2018; Foster & Rakhlin, 2020). The nature of the oracle-efficient guarantees for the contextual bandit problem is much weaker compared to full-information online learning: positive results either assume a stochastic probability model on the responses given covariates (e.g. Foster et al. (2018); Foster & Rakhlin (2020)) or significantly stronger oracles than Eq. (1) (e.g. Agarwal et al. (2014)).

## 2.2 Adaptive online learning

In this paper, we focus on designing oracle-efficient algorithms with *problem-dependent* regret guarantees. Note that this kind of bound can be achieved by many inefficient algorithms in general, such as Hedge and its variants (Cesa-Bianchi & Lugosi, 2006; De Rooij et al., 2014; Luo & Schapire, 2015), follow-the-perturbed-leader (Kalai & Vempala, 2005; Van Erven et al., 2014) or follow-the-regularized-leader (Orabona, 2019). Small-loss bounds can also be obtained efficiently when the loss functions are simply linear (Hutter & Poland, 2005; Syrgkanis et al., 2016). On the other hand, in online convex optimization, small-loss bounds can be obtained when the loss functions are additionally smooth (Srebro et al., 2010; Orabona et al., 2012; Wang et al., 2020). However, these algorithms heavily rely on the special structure of the loss functions. In this paper, we take the first step to extend these methods to support the more complicated (generally non-convex) problems which appear in real-world applications.

Apart from the small-loss, there exist other types of problem-dependent bounds, such as second-order bound (Cesa-Bianchi et al., 2005; Gaillard et al., 2014), quantile bound (Chaudhuri et al., 2009; Koolen & Erven, 2015), or parameter-free bound (Luo & Schapire, 2015; Cutkosky & Orabona, 2018). Moreover, advanced adaptive results can also be obtained by minimizing more advanced performances measures other than regret, such as adaptive regret (Hazan & Seshadhri, 2007; Zhang et al., 2019), or dynamic regret (Zhang et al., 2018; Zhao et al., 2020). How to obtain these more refined theoretical guarantees in the oracle-efficient setting remains an interesting open problem.

## 3  GFTPL with Small-Loss Bound

In this section, we ignore computational complexity for the moment and we provide a new FTPL-type algorithm that enjoys the small-loss bound. We then show that the proposed algorithm can be implemented efficiently by the offline oracle in Section 4. Before diving into the details, we first briefly recall the definition of online learning and regret.

**Preliminaries.**  The online decision problem we consider can be described as follows. In each round $t$, a learner picks an action $x_t \in \mathcal{X} = [x^{(1)}, \ldots, x^{(K)}]$. After observing the adversary's decision $y_t \in \mathcal{Y}$, the learner suffers a loss $f(x_t, y_t)$ where the loss function $f : \mathcal{X} \times \mathcal{Y} \mapsto [0, 1]$ is known to the learner and adversary. The regret of an online learning algorithm $\mathcal{A}$ is defined as

$$R_T^{\mathcal{A}} := \mathbb{E}\left[\sum_{t=1}^T f(x_t, y_t) - L_T^*\right],$$

where $L_T^* = \min_{k \in [K]} \sum_{j=1}^T f(x^{(k)}, y_j)$ is the cumulative loss of the best action in hindsight, and the expectation is taken only with respect to the potentially randomized strategy of the learner.

Our proposed algorithm follows the framework of GFTPL (Dudik et al., 2020). We first briefly introduce to the intuition behind this method. Specifically, in each round $t$, GFTPL picks $x_t$ by solving the following optimization problem:

$$x_t = \underset{k \in [K]}{\operatorname{argmin}} \sum_{j=1}^{t-1} f(x^{(k)}, y_j) + \left\langle \Gamma^{(k)}, \alpha \right\rangle,$$

where $\alpha$ is a $N$-dimensional noise vector ($N \ll K$) generated from a uniform distribution, and $\Gamma^{(k)}$ is the $k$-th row of a matrix $\Gamma \in [0, 1]^{K \times N}$, which is referred to as the *perturbation translation matrix* (PTM). Compared to vanilla FTPL, which generates $K$ random variables (one for each expert),

GFTPL only generates $N$ random variables, where $N$ is much smaller than $K$. Each expert is perturbed by a different linear combination of these random variables based on the PTM $\Gamma$. The results of Dudík et al. (2020) rely on the following assumption on $\Gamma$.

**Definition 1.** ($\delta$-admissibility (Dudík et al., 2020)) *Let $\Gamma \in [0,1]^{K \times N}$ be a matrix, and denote $\Gamma^{(k)}$ as the $k$-th row of $\Gamma$, and $\Gamma^{(k,i)}$ the $i$-th element of $\Gamma^{(k)}$. Then, $\Gamma$ is $\delta$-admissible if (a) $\forall k, k' \in [K]$, $\exists i \in [N]$, such that $\Gamma^{(k,i)} \neq \Gamma^{(k',i)}$; and (b) $\forall i \in [N], k, k' \in [K]$, such that $\Gamma^{(k,i)} \neq \Gamma^{(k',i)}$, then $|\Gamma^{(k,i)} - \Gamma^{(k',i)}| \geq \delta$.*

The $\delta$-admissibility guarantees that every two rows in $\Gamma$ are significantly distinct. As pointed out by Dudík et al. (2020), this is the essential property required by GFTPL, and is used to *stabilize* the algorithm in the analysis, i.e., ensuring that $\mathbb{P}[x_t \neq x_{t+1}]$ is small. However, the adaptive analysis of inefficient FTPL (Hutter & Poland, 2005) (i.e. using a noise vector of dimension equal to the size of the decision set) reveals that this type of stability is insufficient. Instead, one needs to control the following $\forall t$ and $\forall i \in [K]$,

$$\frac{\mathbb{P}[x_t = x^{(i)}]}{\mathbb{P}[x_{t+1} = x^{(i)}]}, \tag{2}$$

the ratio of the probability of picking the $i$-th decision in two consecutive rounds. We note that $\delta$-admissibility is not sufficient to ensure this quantity is bounded, as we establish in the following counter-example lemma. (See Appendix A.1 for proof).

**Lemma 1.** *There is an instance of a $\delta$-admissible $\Gamma$, and a sequence $\{y_t : t = 1, 2, \ldots\}$, such that if we run GFTPL we can have $\frac{\mathbb{P}[x_t = x^{(i)}]}{\mathbb{P}[x_{t+1} = x^{(i)}]} = \infty$ for some $i \in [K]$ and some $t > 0$.*

To address this problem, we propose a new property for $\Gamma$. Define $B_\gamma^1 := \{s \in \mathbb{R}^N : \|s\|_1 \leq \gamma\}$ as the $\ell_1$-ball of size $\gamma$.

**Definition 2.** ($\gamma$-approximability) *Let $\Gamma \in [0,1]^{K \times N}$. We say that $\Gamma$ is $\gamma$-approximable if*

$$\forall k \in [K], y \in \mathcal{Y} \, \exists s \in B_\gamma^1 \, \forall j \in [K] : \qquad \left\langle \Gamma^{(k)} - \Gamma^{(j)}, s \right\rangle \geq f(x^{(k)}, y) - f(x^{(j)}, y).$$

It may not be immediately obvious how we arrived at this condition, so let us provide some intuition. The goal of perturbation methods in sequential decision problems, going back to the early work of Hannan (1957), is to ensure that the algorithm is "hedging" across all available alternative decisions. A newly observed data point $y$ may make expert $j$ suddenly look more attractive than expert $k$, as we have now introduced a new gap $f(x^{(k)}, y) - f(x^{(j)}, y)$ in their measured loss values. With this in mind, we say that $\Gamma$ is a "good" (i.e. approximable) choice for the PTM, if this gap can be overcome (hedged) by some small (i.e. likely) perturbation $s$, so that $\left\langle \Gamma^{(k)} - \Gamma^{(j)}, s \right\rangle$ makes up the difference. The *inequality* makes this property flexible and much easier to satisfy in real-world applications: we only need the gap approximation from above. Later, we will show that $\gamma$-approximability guarantees the required stability measure in (2), and thus is critical for the small-loss bound.

We want to emphasize two final points. First, the $\gamma$-approximability condition is *purely for analysis purposes* and we don't need compute the quantity $s$ in response to $y$ and $k$. Second, much of the computational and decision-theoretic challenges rest heavily on the careful design of $\Gamma$. The PTM allows the algorithm to perform the appropriate hedging across an exponentially-sized set of $K$ experts with only $N \ll K$ dimensions of perturbation. As we demonstrate in the following example, we can always construct a $\gamma$-approximable $\Gamma$, with $N = O(\log K)$, but at the expense of computational efficiency. The proposed $\Gamma$ will not generally be *compatible* with the given oracle, in the sense that the optimization problem underlying GFTPL cannot be written in the form of Eq. (1). In the next section, we will show how to address this problem via another condition on $\Gamma$ called *implementablity*.

**Simple Example** For any online learning problem we may construct $\Gamma$ as follows. Let $N := \lceil \log_2 K \rceil$, and define the $k$th row $\Gamma^{(k)}$ to be the *binary representation* of the index $k$, with $+1/-1$ values instead of $0/1$. We claim that this $\Gamma$ is $\gamma$-approximable, for $\gamma = \lceil \log_2 K \rceil$. We can satisfy the condition of Definition 2, by setting $s = \Gamma^{(k)}$. It is easy to see that for any $j \neq k$ we have $\left\langle \Gamma^{(k)} - \Gamma^{(j)}, s \right\rangle = \left\langle \Gamma^{(k)} - \Gamma^{(j)}, \Gamma^{(k)} \right\rangle \geq 2 \geq f(x^{(k)}, y) - f(x^{(j)}, y)$, where the last inequality holds because $|f(x^{(i)}, y)| \leq 1$ for any $i \in [K]$.

---

**Algorithm 1** Generalized follow-the-perturbed-leader with small-loss bound

---

1: **Init:** $\Gamma \in [0,1]^{K \times N}$
2: Draw IID vector $\alpha = [\alpha^{(1)}, \ldots, \alpha^{(N)}] \sim \mathrm{Lap}(1)^N$; that is, $p(\alpha^{(i)}) = \frac{1}{2} \exp(-|\alpha^{(i)}|)$
3: **for** $t = 1, \ldots, T$ **do**
4:    Set $\alpha_t \leftarrow \frac{\alpha}{\eta_t}$, where $\eta_t > 0$ a parameter computed online
5:    Choose $x_t \leftarrow \operatorname*{argmin}\limits_{k \in [K]} \sum\limits_{j=1}^{t-1} f(x^{(k)}, y_j) + \left\langle \Gamma^{(k)}, \alpha_t \right\rangle$
6:    Observe $y_t$
7: **end for**

---

**Comparison beteeen $\gamma$-approximabilty (this paper) and $\delta$-admissibility (Dudík et al., 2020)**
We note that, although $\gamma$-approximability leads to a much tighter bound, it is not stronger than $\delta$-admissibility. Instead, they are incomparable conditions. Specifically:

- In Section 4.1 we demonstrate that when $\Gamma$ is binary, admissibility directly leads to approximability. As shown by Dudík et al. (2020), a binary and admissible $\Gamma$ exists in various online auctions problems, including VCG with bidder-specific reserves (Roughgarden & Wang, 2019), envy-free item pricing (Guruswami et al., 2005), online welfare maximization in multi-unit auction (Dobzinski & Nisan, 2010), and simultaneous second-price auctions (Daskalakis & Syrgkanis, 2016). We can directly obtain an approximable $\Gamma$ in such cases.

- On the other hand, in problems such as level auction (Dudík et al., 2020), one can construct both admissible and approximable $\Gamma$, although in completely different ways; we discuss the construction in depth in Section 4.1.

- In section 4.2, we show that, when the adversary's action space is small, we can always construct a $\gamma$-approximable $\Gamma$, while a $\delta$-admissible $\Gamma$ does not exist in general.

- In Appendix A.2, we show that in some cases a $\delta$-admissible $\Gamma$ can be obtained while $\gamma$-approximability cannot be achieved.

Equipped with the $\gamma$-approximable PTM, we develop a generalized follow-the-perturbed-leader algorithm with the Laplace distribution for the noise $\alpha$[1] and a time-varying step size, which is summarized in Algorithm 1. This choice of Laplace distribution is significantly different from the choice of uniform distribution originally used by GTFPL: it turns out that a continuous distribution is required to satisfy Eq. (2) and thereby the small-loss bound. Note that here we ignored the time complexity and only focus on the regret. We will specify how to construct $\Gamma$ in the next section. For the proposed algorithm, we successfully obtain the following stronger stability property.

**Lemma 2.** *Assume $\Gamma$ is $\gamma$-approximable. Let $x_t' = \operatorname{argmin}_{k \in [K]} \sum_{j=1}^{t} f(x^{(k)}, y_j) + \langle \Gamma^{(k)}, \alpha_t \rangle$. Then in each round $t$, we have $\forall i \in [K]$,*

$$\mathbb{P}[x_t = x^{(i)}] \leq \exp\left(\gamma \eta_t\right) \mathbb{P}[x_t' = x^{(i)}].$$

Note that we replace the term $x_{t+1}$ in (2) with $x_t'$, as a time-varying step-size is used. Based on Lemma 2, we obtain the regret bound of Algorithm 1 as follows.

**Theorem 1.** *Assume $\Gamma$ is $\gamma$-approximable, and let $L_T^* = \min_{k \in [K]} \sum_{j=1}^{T} f(x^{(k)}, y_j)$. Algorithm 1, with $\eta_t = \min\left\{\frac{1}{\gamma}, \frac{c}{\sqrt{L_{t-1}^* + 1}}\right\}$ for any $c > 0$, achieves the following regret bound:*

$$
R_T \leq \left( \frac{4\sqrt{2} \max\{2 \ln K, \sqrt{N \ln K}\}}{c} + 2\gamma \left( c + \frac{1}{c} \right) \right) \sqrt{L_T^* + 1}
$$
$$
+ 8\gamma \ln \left( \frac{1}{c} \sqrt{L_T^* + 1} + \gamma \right) + 2\gamma^2 + 4\sqrt{2} \max\{2 \ln K, \sqrt{N \ln K}\} \gamma. \tag{3}
$$

---

[1] Note that the Laplace distribution is not the unique choice to get the small-loss bound. In Appendix A.5, we prove that the $\ell_p$ perturbation $p(\alpha) \propto \exp\left\{ -\left( \sum_i |\alpha^{(i)}|^p \right)^{\frac{1}{p}} \right\}$ indeed works for any $p \geq 1$.

---
**Algorithm 2** Oracle-based GFTPL for the reward feedback
---
1: **Input:** Data set $S_j$, $j \in [N]$, that implement a matrix $\Gamma \in [0,1]^{K \times N}$, $\eta_1 = \min\{\frac{1}{\gamma}, 1\}$.
2: Draw IID vector $\alpha = [\alpha^{(1)}, \ldots, \alpha^{(N)}] \sim \mathrm{Lap}(1)^N$
3: **for** $t = 1, \ldots, T$ **do**
4:    Choose $x_t \leftarrow \operatorname*{argmin}_{k \in [K]} \sum_{j=1}^{t-1} f(x^{(k)}, y_j) + \sum_{i=1}^{N} \frac{\alpha^{(i)}}{\eta_t} \left[ \sum_{(w,y) \in \mathcal{S}_i} w \cdot r(x^{(k)}, y) \right]$
5:    Observe $y_t$
6:    Compute $\widehat{L}_t^* = \min_{k \in [K]} \sum_{j=1}^{t} f(x^{(k)}, y_j)$ by using the oracle
7:    Set $\eta_{t+1} \leftarrow \min \left\{ \frac{1}{\gamma}, \frac{1}{\sqrt{\widehat{L}_t^* + 1}} \right\}$
8: **end for**
---

The proof of Lemma 2 and Theorem 1 can be found in Appendix A.3. By setting $c = \Theta(1)$, Theorem 1 implies that our proposed algorithm achieves $O(\max\{\gamma, \ln K, \sqrt{N \ln K}\}\sqrt{L_T^*})$ regret bound.

**Comparison to GFTPL (Dudík et al., 2020)**   The original GFTPL algorithm has an $O(\frac{N}{\delta}\sqrt{T})$ regret bound. For the dependence on $T$, our $O(\sqrt{L_T^*})$ bound reduces to $O(\sqrt{T})$ in the worst-case, and automatically becomes tighter when $L_T^*$ is small. On the other hand, for the dependence on other terms, we note that both $\frac{N}{\delta}$ and $\max\{\gamma, \ln K, \sqrt{N \ln K}\}$ are lower bounded by $\Omega(\ln K)$, and their exact relationship depends on the specific problem. In Section 4, we show that for many auction applications, the two terms are on the same order. Moreover, in cases such as when $|\mathcal{Y}|$ is small, Algorithm 1 with an appropriate $c$ leads to $O\left(\sqrt{L_T^* \max\{\ln K, \sqrt{|\mathcal{Y}| \ln K}\}}\right)$ regret bound, while the regret bound of GFTPL in Dudík et al. (2020) can blow up since $\delta$ can be infinitely small.

## 4   Oracle-efficiency and Applications

In this section, we discuss how to run Algorithm 1 in an oracle-efficient way. Following Dudík et al. (2020), we introduce the following definition.

**Definition 3** (Implementability). *A matrix $\Gamma$ is implementable with complexity $M$ if for each $j \in [N]$ there exists a dataset $S_j$, with $|S_j| \leq M$, such that $\forall k, k' \in [K]$,*

$$\Gamma^{(k,j)} - \Gamma^{(k',j)} = \sum_{(w,y) \in \mathcal{S}_j} w \left( f(x^{(k)}, y) - f(x^{(k')}, y) \right).$$

Based on Definition 3, it is easy to get the following theorem, which is similar to Theorem 2.10 of Dudík et al. (2020).

**Theorem 2.** *If $\Gamma$ is implementable, then Algorithm 1 is oracle-efficient and has a per-round complexity $O(T + NM)$.*

In the following sub-sections, we discuss how to construct approximable and implementable $\Gamma$ matrices in different applications.

### 4.1   Applications in online auctions

In this part, we apply Algorithm 1 to online auction problems, which is the main focus of Dudík et al. (2020). To deal with this sort of problems, we first transform Algorithm 1 to online learning with rewards setting, i.e., in each round $t$, after choosing $x_t$, instead of suffering a loss, the learner obtains a reward $r(x_t, y_t) \in [0,1]$. For this case, it is straightforward to see that running Algorithm 1 on a surrogate loss $f(x, y) = 1 - r(x, y)$ directly leads to the small-loss bound. To proceed, we slightly change this procedure and obtain Algorithm 2. The main difference is that, we implement $\Gamma$ with the reward function $r(x, y)$, instead of the surrogate loss $f(x, y)$. This makes the construction of $\Gamma$ much easier. We have the following regret bound for Algorithm 2.

**Corollary 1.** *Let $f(x,y) = 1 - r(x,y)$. Assume $\Gamma$ is $\gamma$-approximable w.r.t. $f(x,y)$ and implementable with function $r(x,y)$. Then Algorithm 2 is oracle-efficient and achieves the following regret bound:*

$$R_T = \mathbb{E}\left[G_T^* - \sum_{t=1}^{T} r(x_t, y_t)\right] = O\left(\max\left\{\gamma, \ln K, \sqrt{N \ln K}\right\}\sqrt{T - G_T^*}\right),$$

*where $G_T^* = \max_{i \in [K]} \sum_{t=1}^{T} r(x^{(i)}, y_t)$ is the cumulative reward of the best expert.*

Next, we discuss how to construct the PTM in several auction problems.

**Auctions with binary and admissible $\Gamma$.**    As shown by Dudík et al. (2020), in many online auction problems, such as the Vickrey-Clarkes-Groves (VCG) mechanism with bidder-specific reserves (Roughgarden & Wang, 2019), envy-free item pricing (Guruswami et al., 2005), online welfare maximization in multi-unit auction (Dobzinski & Nisan, 2010) and simultaneous second-price auctions (Daskalakis & Syrgkanis, 2016), there exists a binary PTM which is 1-admissible and implementable with $N$ rows where $N \ll K$. For these cases, we have the following lemma. The proof is deferred to Appendix B.1.

**Lemma 3.** *Let $\Gamma \in [0,1]^{K \times N}$ be a binary matrix and 1-admissible, then $\Gamma$ is $N$-approximable.*

Note that, $\Gamma$ is binary and 1-admissible, so every two rows of Gamma differ by at least one element. This means that $\Gamma$ must, at the very least, include $\Omega(\ln K)$ columns to encode each row. Combining this fact with Lemma 3 and Corollary 1, we can obtain an $O(N\sqrt{T - L_T^*})$ bound for all of the above problems. Compared to the original GFTPL algorithm, our condition leads to a similar dependence on $N$ and a tighter dependence on $T$ due to the improved small-loss bound. More details about the aforementioned auction problems and corresponding regret bounds can be found in Appendix B.2.

**Level auction**    The class of level auctions was first introduced by Morgenstern & Roughgarden (2015), and optimizing over this class enables a $(1 - \epsilon)$ multiplicative approximation with respect to Myerson's optimal auction when the distribution of each bidder's valuation is independent from others. For this problem, the PTM in Dudík et al. (2020) is not easily to be shown approximable. To address this problem, we propose a novel way of constructing an approximable and implementable PTM. *The key idea is to utilize a coordinate-wise threshold function to implement $\Gamma$.* Note that this kind of function can not be directly obtained. Instead, we create an augmented problem with a surrogate loss to deal with this issue. For level auction with single-item, $n$-bidders, $s$-level and $m$-discretization level, our method enjoys an $O(nsm\sqrt{T - L_T^*})$ regret bound, which is tighter than the $O(nm^2\sqrt{T})$ (*note that $s \leq m$*) of the original GFTPL both on its dependence on the number of rounds $T$ and auction parameters $n, s, m$. Due to page limitations, we postpone the detailed problem description and proof to Appendix B.3.

## 4.2   Other applications

**Oracle learning and finite parameter space**    In many real-world applications, such as security game (Balcan et al., 2015) and online bidding with finite threshold vectors (Daskalakis & Syrgkanis, 2016), the decision set $\mathcal{X}$ is extremely large, while the adversary's action set $|\mathcal{Y}|$ is finite and small. For these problems, we can construct an implementable PTM based the following lemma, whose proof can be found in Appendix B.4.

**Lemma 4.** *Consider the setting with $|\mathcal{Y}| = d$ ($d \ll K$), then there exists a 1-approximable and implementable $\Gamma$ with $d$ columns and complexity 1.*

Combining Lemma 4 and Theorem 1, and configuring $c = \sqrt{\max\{\ln K, \sqrt{d \ln K}\}}$, we observe that our algorithm achieves a small-loss bound on the order of $O(\sqrt{\max\{\ln K, \sqrt{d \ln K}\}}\sqrt{L_T^*})$. On the other hand, because of the continuity of the loss functions in this setting, a $\delta$-admissible PTM in general does not exist (as $\delta$ may approach 0). Therefore, our proposed condition not only leads to a tighter bound, but can also solve problems that the original GFTPL (Dudík et al., 2020) can not handle.

**Transductive online classification**  Finally, we consider the transductive online classification problem (Syrgkanis et al., 2016; Dudík et al., 2020). In this setting, the decision set $\mathcal{X}$ consists of $K$ binary classifiers. In each round $t$, firstly the adversary picks a feature vector $w_t \in \mathcal{W}$, where $|\mathcal{W}| = m$. Then, the learner chooses a classifier $x_t(\cdot)$ from $\mathcal{X}$. After that, the adversary reveals the label $y_t \in \{0, 1\}$, and the learner suffers a loss $f(x_t, (w_t, y_t)) = \mathbb{I}[x_t(w_t) \neq y_t]$. We assume the problem is transductive, i.e., the learner has access to the adversary's set of vectors at the beginning. For this setting, we achieve the following results (the proof is in Appendix B.5).

**Lemma 5.** *Consider transductive online classification with $|\mathcal{W}| = m$. Then there exists a 1-approximable and implementable PTM with $m$ columns and complexity 1. Moreover, Algorithm 1 with such a PTM and appropriately chosen parameters achieves $O(\sqrt{\max\{\ln K, \sqrt{m \ln K}\}} \sqrt{L_T^*})$ regret.*

**Negative implementability**  In the this paper we assume that the offline oracle can solve the minimization problem in (1) given any real-weights. In some cases, the oracle can only accept positive weights. This problem can be solved by constructing negative implementable PTM (Dudík et al., 2020). In most of the cases discussed above, negative implementable and approximable PTM exist. This is formally shown in Appendix B.6.

## 5  Best-of-Both-Worlds Bound: Adapting to IID data

In this section, we switch our focus to adapting between adversarial and stochastic data. While the GFTPL algorithm enjoys an $O(\sqrt{L_T^*})$-type regret bound on adversarial data, it is possible to obtain much better rates on stochastic data. For example, by setting all step sizes $\eta_t$ as $\infty$, Algorithm 1 reduces to the classical FTL algorithm, which suffers linear regret in the adversarial setting but enjoys much tighter bounds when the data is IID or number of leader changes is small. To be more specific, we introduce the following regret bound for FTL.

**Lemma 6** (Lemma 9, De Rooij et al. (2014)). *Let $x_t^{\mathsf{FTL}} = \arg\min_{i \in [K]} \sum_{s=1}^{t-1} f(x^{(i)}, y_s)$ be the output of the FTL algorithm at round $t$, $C_T$ the set of rounds where the leader changes, and $\delta_t = f(x_t^{\mathsf{FTL}}, y_t) - (L_t^* - L_{t-1}^*)$ the "mixability gap"[2] at round $t$. Then for any $T \geq 1$, the regret of FTL is bounded by $R_T^{\mathsf{FTL}} \leq \sum_{t \in C_T} \delta_t \leq |C_T|$.*

Note that since $f \in [0, 1]$ and $L_t^* - L_{t-1}^* \in [0, f(x_t^{\mathsf{FTL}}, y_t)]$, we know $\delta_t \in [0, 1]$. For the *i.i.d* case, if the mean loss of the best expert is smaller than that of other experts by a constant, then due to the law of large numbers, the number of leader changes would be small, which results in a constant regret bound (De Rooij et al., 2014).

Our goal is to obtain a "best-of-both-worlds" bound, which can ensure the small-loss bound in general, while automatically leading to tighter bounds for IID data like FTL. We will now design an algorithm that achieves such a bound by adaptively choosing between GFTPL and FTL depending on which algorithm appears to be achieving a lower regret. The essence of this idea was first introduced in the FlipFlop algorithm (De Rooij et al., 2014), who showed best-of-both-worlds bounds in the inefficient case. Our contribution in this section is to adapt this idea to the oracle-efficient setting. Denote $U_T^{\mathsf{GFTPL}}$ as the attainable regret bound (as in Theorem 1) for running Algorithm 1 alone and $U_T^{\mathsf{FTL}} = \sum_{t \in C_T} \delta_t$ to be that of FTL. In the following, we develop a new algorithm and prove that it is optimal in both worlds, that is, its regret is on the order of $O(\min\{U_T^{\mathsf{FTL}}, U_T^{\mathsf{GFTPL}}\})$.

The proposed algorithm, named as oracle-efficient flipflop (OFF) algorithm, is summarized in Algorithm 3. The core idea is to switch between FTL and GFTPL (Algorthm 1) based on the comparison of the estimated regret. We optimistically start from FTL. In each round $t$, we firstly pick $x_t$ based on the current algorithm $\mathsf{Alg}_t$, and then obtain the adversary's action $y_t$ (line 2). Next, we compute the estimated bounds of regret of both algorithms until round $t$ (line 3). Specifically, let $\mathcal{I}_t^{\mathsf{FTL}} = \{i | i \in [t], \mathsf{Alg}_i = \mathsf{FTL}\}$ and $\mathcal{I}_t^{\mathsf{GFTPL}} = \{i | i \in [t], \mathsf{Alg}_i = \mathsf{GFTPL}\}$ be the set of rounds up to $t$ in which we run FTL and GFTPL. Then, the estimated regret of FTL in $\mathcal{I}_t^{\mathsf{FTL}}$ is given by

---

[2]Here, we use the special definition of the mixability gap for the FTL algorithm. The details can be found in the second paragraph, page 1286 of De Rooij et al. (2014).

---

**Algorithm 3** Oracle-efficient Flipflop (OFF)

---
**Initialization:** $\mathsf{Alg}_1 = \mathsf{FTL}$

1: **for** $t = 1, \ldots, T$ **do**
2:      Get $x_t$ by $\mathsf{Alg}_t$, observe $y_t$
3:      Compute $\widehat{U}_t^{\mathsf{FTL}}$ and $\widehat{U}_t^{\mathsf{GFTPL}}$
4:      **if** $\mathsf{Alg}_t == \mathsf{FTL}$ and $\widehat{U}_t^{\mathsf{FTL}} > \alpha \widehat{U}_t^{\mathsf{GFTPL}}$ **then**
5:         $\mathsf{Alg}_{t+1} = \mathsf{GFTPL}$
6:      **else if** $\mathsf{Alg}_t == \mathsf{GFTPL}$ and $\widehat{U}_t^{\mathsf{GFTPL}} > \beta \widehat{U}_t^{\mathsf{FTL}}$ **then**
7:         $\mathsf{Alg}_{t+1} = \mathsf{FTL}$
8:      **end if**
9:      Feed $y_t$ to $\mathsf{Alg}_{t+1}$
10: **end for**

---

$\widehat{U}_t^{\mathsf{FTL}} = \sum_{i \in \mathcal{I}_t^{\mathsf{FTL}}} \delta_i$, and the estimated regret of GFTPL in $\mathcal{I}_t^{\mathsf{GFTPL}}$ can be bounded via Theorem 1:

$$
\begin{aligned}
\widehat{U}_t^{\mathsf{GFTPL}} = {}& \left( 4\sqrt{2} \max\{2 \ln K, \sqrt{N \ln K}\} + 4\gamma \right) \sqrt{\widehat{L}_t^* + 1} \\
& + 8\gamma \ln \left( \sqrt{\widehat{L}_t^* + 1} + \gamma \right) + 2\gamma^2 + 4\sqrt{2} \max\{2 \ln K, \sqrt{N \ln K}\}\gamma.
\end{aligned} \tag{4}
$$

where $\widehat{L}_t^* = \min_{x \in \mathcal{X}} \sum_{i \in \mathcal{I}_t^{\mathsf{GFTPL}}} f(x, y_i)$ and we set $c = 1$. Note that, the two quantities defined above are the exact regret upper bounds of the two algorithms on their sub-time intervals up to round $t$, due to the fact that the regret bounds provided in Lemma 6 and Theorem 1 are *timeless*. Moreover, note that the two values can be computed by the oracle. We compare the estimated regret of both algorithms, and use the algorithm which performs better for the next round (lines 4-8).

For the proposed algorithm, we have the following theoretical guarantee (the proof can be found in Appendix C).

**Theorem 3.** *Assume we have a $\gamma$-approximable $\Gamma$, then Algorithm 3 is able to achieve the following bound:*
$$
R_T^{OFF} \le \min \left\{ 3U_T^{GFTPL} + 1, 3U_T^{FTL} + \tau \right\},
$$
*where $\tau = 4\sqrt{2} \max\{2 \ln K, \sqrt{N \ln K}\} + 12\gamma$ and $\alpha = \beta = 1$.*

The Theorem above shows that the regret of Algorithm 3 is the minimum of the regret upper bounds of GFTPL and FTL. Thus, it ensures the $O(\sqrt{T})$-type bound in the worst case, while automatically achieves the much better constant regret bound of FTL under iid data without knowing the presence of stochasticity in data beforehand.

## 6   Conclusion

In this paper, we establish a sufficient condition for the first-order bound in the oracle-efficient setting by investigating a variant of the generalized follow-the-perturbed-leader algorithm. We also show the condition is satisfied in various applications. Finally, we extend the algorithm to adapt to IID losses and achieve a "best-of-both-worlds" bound. In the future, we would like to investigate how to achieve tighter results for oracle-efficient setting, such as the second-order bound (De Rooij et al., 2014) and the quantile bound (Koolen & Erven, 2015).

**Acknowledgments.** We gratefully thank the AI4OPT Institute for funding, as part of NSF Award 2112533. We gratefully acknowledge the NSF for their support through Award IIS-2212182 and Adobe Research for their support through a Data Science Research Award. Part of this work was conducted while the authors were visiting the Simons Institute for the Theory of Computing.

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
