# A Omitted Proofs from Section 3

In this section, we provide the omitted proofs from Section 3.

## A.1 Proof of Lemma 1

Recall that in GFTPL (Dudík et al., 2020), $x_t$ is picked by solving the following optimization problem:

$$x_t = \operatorname{argmin}_{k \in [K]} \sum_{s=1}^{t-1} f(x^{(k)}, y_s) + \langle \Gamma^{(k)}, \alpha \rangle,$$

where the entries of the random vector $\alpha$ are sampled from a uniform distribution $\mathcal{U}[0, \beta]^N$ for some hyperparameter $\beta > 0$. Recall that to prove Lemma 1, we wish to find a counterexample of a $\delta$-admissible PTM $\Gamma$ and a sequence $\{y_t : t = 1, 2, \dots, \}$ such that the probability distribution induced by GFTPL yields $\frac{\mathbb{P}[x_t = x^{(i)}]}{\mathbb{P}[x_{t+1} = x^{(i)}]} = \infty$ for some $i \in [K]$ and some $t > 0$. This precludes the possibility of obtaining a stronger small-loss bound on instances that only satisfy $\delta$-admissibility and no other special properties.

In the following, we show that such a counterexample can be found in this case by exploiting the property of the bounded support of the distribution. We then demonstrate that a similar counterexample exists even if the uniform distribution is replaced by distributions with unbounded support.

### A.1.1 Case 1: Uniform noise distribution (or, more generally, distributions with bounded support)

Fix $K = 2$ experts and round $t > 0$. Suppose that we can obtain an implementable $\Gamma = [0, 1]^\top$ which is 1-admissible with $N = 1$ column. In this case, $\alpha$ is a scalar random variable, and we have

$$\mathbb{P}[x_t = x^{(1)}] = \mathbb{P}\left[\sum_{s=1}^{t-1} f(x^{(1)}, y_s) \leq \sum_{s=1}^{t-1} f(x^{(2)}, y_s) + \alpha\right]$$

$$= \mathbb{P}\left[\alpha \geq \sum_{s=1}^{t-1} f(x^{(1)}, y_s) - \sum_{s=1}^{t-1} f(x^{(2)}, y_s)\right].$$

Similarly, at round $t + 1$ we have

$$\mathbb{P}[x_{t+1} = x^{(1)}] = \mathbb{P}\left[\alpha \geq \sum_{s=1}^{t} f(x^{(1)}, y_s) - \sum_{s=1}^{t} f(x^{(2)}, y_s)\right].$$

Since the probability density function of $\alpha$ has bounded support, it is straightforward to pick appropriate loss functions such that $\sum_{s=1}^{t} f(x^{(1)}, y_s) - \sum_{s=1}^{t} f(x^{(2)}, y_s)$ lies outside the support of the density function while $\sum_{s=1}^{t-1} f(x^{(1)}, y_s) - \sum_{s=1}^{t-1} f(x^{(2)}, y_s)$ lies inside the support of the density function. As a consequence, we get $\mathbb{P}[x_{t+1} = x^{(1)}] = 0$ while $\mathbb{P}[x_t = x^{(1)}] > 0$.

### A.1.2 Case 2: Noise distributions with unbounded support

One may argue that the above bad case happens mainly because the noise density has bounded support. We now show that such counterexamples can also be constructed when the noise $\alpha$ is generated from distributions with unbounded support, such as the Laplace distribution — with a slightly larger number of experts. Specifically, we consider $K = 3$ experts and the PTM $\Gamma = [0, 0.5, 1]^\top$, which is 0.5-admissible with $N = 1$ column. Then, we have

$$\mathbb{P}\left[x_t = x^{(2)}\right]$$

$$= \mathbb{P}\left[\sum_{s=1}^{t-1} f(x^{(2)}, y_s) + 0.5\alpha \leq \sum_{s=1}^{t-1} f(x^{(1)}, y_s) \text{ and } \sum_{s=1}^{t-1} f(x^{(2)}, y_s) + 0.5\alpha \leq \sum_{s=1}^{t-1} f(x^{(3)}, y_s) + \alpha\right]$$

$$= \mathbb{P}\left[2\left(\sum_{s=1}^{t-1} f(x^{(2)}, y_s) - \sum_{s=1}^{t-1} f(x^{(3)}, y_s)\right) \leq \alpha \leq 2\left(\sum_{s=1}^{t-1} f(x^{(1)}, y_s) - \sum_{s=1}^{t-1} f(x^{(2)}, y_s)\right)\right]$$

$$= \mathbb{P}\left[2\Delta_{23} \leq \alpha \leq 2\Delta_{12}\right],$$

where we have defined $\Delta_{23} = \sum_{s=1}^{t-1} f(x^{(2)}, y_s) - \sum_{s=1}^{t-1} f(x^{(3)}, y_s)$, and $\Delta_{12} = \sum_{s=1}^{t-1} f(x^{(1)}, y_s) - \sum_{s=1}^{t-1} f(x^{(2)}, y_s)$ as shorthand. Now, we pick loss functions such that $\Delta_{12} > \Delta_{23}$ and $\Delta_{12} - \Delta_{23} = \epsilon < 1$. Because $\Delta_{12} > \Delta_{23}$ and the distribution of $\alpha$ has infinite support, we have $\mathbb{P}[x_t = x^{(2)}] > 0$. On the other hand, for round $t+1$ a similar argument yields

$$\mathbb{P}[x_{t+1} = x^{(2)}]$$
$$= \mathbb{P}\left[2\Delta_{23} + 2\left(f(x^{(2)}, y_t) - f(x^{(3)}, y_t)\right) \le \alpha \le 2\Delta_{12} + 2\left(f(x^{(1)}, y_t) - f(x^{(2)}, y_t)\right)\right].$$

Now, we pick $f(x^{(2)}, y_t) = 0.5$, and $f(x^{(1)}, y_t) = f(x^{(3)}, y_t) = 0$. For this choice, we get $2\Delta_{23} + 2\left(f(x^{(2)}, y_t) - f(x^{(3)}, y_t)\right) \ge 2\Delta_{12} + 2\left(f(x^{(1)}, y_t) - f(x^{(2)}, y_t)\right)$ which implies that $\mathbb{P}[x_{t+1} = x^{(2)}] = 0$. This in turn implies that $\frac{\mathbb{P}[x_t = x^{(2)}]}{\mathbb{P}[x_{t+1} = x^{(2)}]} = \infty$, completing the proof of the counterexample. $\qquad\square$

These counterexamples imply that the condition of $\delta$-admissibility alone on the PTM $\Gamma$ is not sufficient to control the stronger stability measure required for a small-loss bound. Consequently, new assumptions on $\Gamma$ need to be introduced.

### A.2 Counterexamples showing that $\delta$-admissiblity does not necessarily lead to $\gamma$-approximability

In this paper, we introduced a new sufficient condition of $\gamma$-*approximability* that implies not only worst-case regret bounds but also regret bounds that adapt to the size of the best loss in hindsight. It is natural to ask about the relationship of this sufficient condition with $\delta$-admissibility. In this section, we show that exist $\delta$-admissible PTMs that do not satisfy $\gamma$-approximability. (Note that the reverse statement is also true: Lemma 4 constructs $\gamma$-approximable PTMs that are not in general $\delta$-admissible.)

The counterexample is precisely the one used in Section A.1.2. That is, there are $K = 3$ experts, and the PTM is given by $\Gamma = [0, 0.5, 1]^\top$. Note that $\Gamma$ is 0.5-admissible with one column. Further, we consider an output $y$ such that $f(x^{(1)}, y) = f(x^{(3)}, y) = 0$ and $f(x^{(2)}, y) = 1$. We proceed to show that this PTM $\Gamma$ is not approximable. To prove this, note that for some scalar $s$ to satisfy the requisite approximability condition, we need $(\Gamma^{(2)} - \Gamma^{(1)})s = 0.5s \ge 1$ *and* $(\Gamma^{(2)} - \Gamma^{(3)})s = -0.5s \ge 1$. This is clearly unsatisfiable by any scalar $s$.

### A.3 Proof of Theorem 1

We now provide the detailed proof of Theorem 1. We begin by introducing some notation specific to this proof. We denote by $\Gamma^{(x_t)}$ the row of $\Gamma$ related to expert $x_t$, and by $\Gamma^*$ the row related to the best-expert-in-hindsight $x^*$. Further, $\Gamma^{(k)}$ denotes the row of $\Gamma$ related to expert $x^{(k)}$ and $\Gamma^{(k,i)}$ denotes the $i$-th component of the row $\Gamma^{(k)}$. We also denote the PDF of the noise vector at round $t$, $\alpha_t$, as $p(\alpha_t)$. Finally, the learner's action set is denoted by $\mathcal{X} = \{x^{(1)}, \ldots, x^{(k)}, \ldots, x^{(K)}\}$.

Our proof begins with the framework used by typical FTPL analyses (Hutter & Poland, 2005; Syrgkanis et al., 2016; Dudík et al., 2020). We first divide the regret into two terms:

$$R_T = \mathbb{E}\left[\sum_{t=1}^{T} f(x_t, y_t) - f(x^*, y_t)\right]$$

$$= \underbrace{\mathbb{E}\left[\sum_{t=1}^{T} f(x_t, y_t) - \sum_{t=1}^{T} f(x_t', y_t)\right]}_{\text{TERM 1}} + \underbrace{\mathbb{E}\left[\sum_{t=1}^{T} f(x_t', y_t) - \sum_{t=1}^{T} f(x^*, y_t)\right]}_{\text{TERM 2}}. \tag{5}$$

Above, the expectation $\mathbb{E}[\cdot]$ is only with respect to the internal randomness of the learner and $x^* = \mathrm{argmin}_{i \in [K]} \sum_{t=1}^{T} f(x^{(i)}, y_t)$ is the best decision in hindsight. Further, the expert

$$x_t' = \mathrm{argmin}_{k \in [K]} \sum_{j=1}^{t} f(x^{(k)}, y_j) + \langle \Gamma^{(k)}, \alpha_t \rangle$$

is usually referred to as the *infeasible* leader (Hutter & Poland, 2005) at round $t$, since $y_t$ can only be obtained after $x_t$ is chosen.

Next, we bound the two terms of (5) respectively. TERM 1 measures the *stability* of GFTPL by tracking how close its performance is to that of the idealized infeasible leader. We obtain the following upper bound on TERM 1 which heavily leverages the key technical Lemma 2.

**Lemma 7.** *Assume that the PTM $\Gamma$ is $\gamma$-approximable, and Algorithm 1 is applied with $\eta_t = \min\left\{\frac{1}{\gamma}, \frac{c}{\sqrt{L_{t-1}^* + 1}}\right\}$, where $L_{t-1}^* = \min_{k \in [K]} \sum_{j=1}^{t-1} f(x^{(k)}, y_j)$ and $c > 0$ is some universal constant. Then for all $T \geq 1$ we have:*

$$\text{TERM 1} \leq \left(\frac{2\sqrt{2}\max\{2\ln K, \sqrt{N \ln K}\}}{c} + 2\gamma\left(c + \frac{1}{c}\right)\right)\sqrt{L_T^* + 1}$$

$$+ 8\gamma \ln\left(\frac{1}{c}\sqrt{L_T^* + 1} + \gamma\right) + 2\gamma^2 + 2\sqrt{2}\gamma \max\{2\ln K, \sqrt{N \ln K}\}.$$

Next, TERM 2 measures the approximation error between the infeasible leader and the true best expert in hindsight. The following lemma, which is a simple extension of the classical be-the-leader lemma (Cesa-Bianchi & Lugosi, 2006), bounds TERM 2.

**Lemma 8.** *Assume that the PTM $\Gamma$ is $\gamma$-approximable. Then, for all $T \geq 1$, we have*

$$\text{TERM 2} \leq 2\sqrt{2}\max\{2\ln K, \sqrt{N \ln K}\}\left(\gamma + \frac{1}{c}\sqrt{L_T^*}\right)$$

We prove Lemmas 2, 7, and 8 in Appendix A.3.1, A.3.2 and A.3.3 respectively. The proof of Theorem 1 follows by directly combining (5), Lemma 7 and Lemma 8.

### A.3.1 Proof of Lemma 2

Note that we can write

$$\mathbb{E}[f(x_t, y_t)] = \sum_{i=1}^{K} f(x^{(i)}, y_t)\mathbb{P}[x_t = x^{(i)}].$$

Our approach will relate $\mathbb{P}[x_t = x^{(i)}]$ and $\mathbb{P}[x_t' = x^{(i)}]$ for every $i \in [K]$: at a high level, a similar approach is also used in the analysis of contextual online learning for linear functions by Syrgkanis et al. (2016) (although several aspects of our analysis are different). Then, for any fixed choice of $s^{(i)} \in \mathbb{R}^N$ we have

$$\mathbb{P}[x_t = x^{(i)}]$$

$$= \int_{\alpha_t} \mathbb{I}\left[\left\{\operatorname*{argmin}_{k \in [K]} \sum_{j=1}^{t-1} f(x^{(k)}, y_j) + \left\langle \Gamma^{(k)}, \alpha_t \right\rangle\right\} = x^{(i)}\right] p(\alpha_t) d\alpha_t$$

$$= \int_{\alpha_t} \mathbb{I}\left[\left\{\operatorname*{argmin}_{k \in [K]} \sum_{j=1}^{t-1} f(x^{(k)}, y_j) + \left\langle \Gamma^{(k)}, \alpha_t \right\rangle\right\} = x^{(i)}\right] p\left(\alpha_t - s^{(i)}\right) \frac{p(\alpha_t)}{p\left(\alpha_t - s^{(i)}\right)} d\alpha_t$$

$$\overset{(1)}{\leq} \sup_{\beta \in \mathbb{R}^N} \frac{p(\beta)}{p\left(\beta - s^{(i)}\right)} \int_{\alpha_t} \mathbb{I}\left[\left\{\operatorname*{argmin}_{k \in [K]} \sum_{k=1}^{t-1} f(x^{(k)}, y_j) + \left\langle \Gamma^{(k)}, \alpha_t \right\rangle\right\} = x^{(i)}\right] p\left(\alpha_t - s^{(i)}\right) d\alpha_t$$

$$\overset{(2)}{\leq} \exp\left(\eta_t \|s^{(i)}\|_1\right) \int_{\alpha_t} \mathbb{I}\left[\left\{\operatorname*{argmin}_{k \in [K]} \sum_{j=1}^{t-1} f(x^{(k)}, y_j) + \left\langle \Gamma^{(k)}, \alpha_t \right\rangle\right\} = x^{(i)}\right] p\left(\alpha_t - s^{(i)}\right) d\alpha_t$$

$$= \exp\left(\eta_t \|s^{(i)}\|_1\right) \int_{\alpha_t} \mathbb{I}\left[\left\{\operatorname*{argmin}_{k \in [K]} \sum_{j=1}^{t-1} f(x^{(k)}, y_j) + \left\langle \Gamma^{(k)}, \alpha_t + s^{(i)} \right\rangle\right\} = x^{(i)}\right] p(\alpha_t) d\alpha_t,$$
$$(6)$$

Above, $\mathbb{I}[\cdot]$ denotes the indicator function and inequality (2) is based on the fact that for any $\beta \in \mathbb{R}^N$,

$$\frac{p(\beta)}{p\left(\beta - s^{(i)}\right)} = \exp\left(\eta_t\left(\left\|\beta - s^{(i)}\right\|_1 - \|\beta\|_1\right)\right) \leq \exp\left(\eta_t \|s^{(i)}\|_1\right), \tag{7}$$

and the final equality is because the support of $\alpha_t$ is unbounded. To proceed, we introduce and prove the following lemma.

**Lemma 9.** *Suppose $\Gamma$ is $\gamma$-approximable. Then, $\forall i \in [N]$ there exists a vector $s^{(i)} \in \mathbb{R}^N$ such that*

$$
\mathbb{I}\left[\left\{\operatorname*{argmin}_{k \in [K]} \sum_{j=1}^{t-1} f(x^{(k)}, y_j) + \left\langle \Gamma^{(k)}, \alpha_t \right\rangle + \left\langle \Gamma^{(k)}, s^{(i)} \right\rangle \right\} = x^{(i)}\right]
$$

$$
\leq \mathbb{I}\left[\left\{\operatorname*{argmin}_{k \in [K]} \sum_{j=1}^{t-1} f(x^{(k)}, y_j) + \left\langle \Gamma^{(k)}, \alpha_t \right\rangle + f(x^{(k)}, y_t) \right\} = x^{(i)}\right]
$$

(8)

*holds for all $\alpha_t$.*

*Proof.* For any fixed $\alpha_t$, if

$$
\mathbb{I}\left[\left\{\operatorname*{argmin}_{k \in [K]} \sum_{j=1}^{t-1} f(x^{(k)}, y_j) + \left\langle \Gamma^{(k)}, \alpha_t \right\rangle + f(x^{(k)}, y_t) \right\} = x^{(i)}\right] = 1,
$$

then the required inequality always holds since the indicator function is upper bounded by 1. For the case when

$$
\mathbb{I}\left[\left\{\operatorname*{argmin}_{k \in [K]} \sum_{j=1}^{t-1} f(x^{(k)}, y_j) + \left\langle \Gamma^{(k)}, \alpha_t \right\rangle + f(x^{(k)}, y_t) \right\} = x^{(i)}\right] = 0,
$$

assume that $x^{(\ell)} = \operatorname*{argmin}_{k \in [K]} \sum_{j=1}^{t-1} f(x^{(k)}, y_j) + \left\langle \Gamma^{(k)}, \alpha_t \right\rangle + f(x^{(k)}, y_t)$ for some $\ell \neq i$. Then

$$
\sum_{j=1}^{t-1} f(x^{(\ell)}, y_j) + \left\langle \Gamma^{(\ell)}, \alpha_t \right\rangle + f(x^{(\ell)}, y_t) \leq \sum_{j=1}^{t-1} f(x^{(i)}, y_j) + \left\langle \Gamma^{(i)}, \alpha_t \right\rangle + f(x^{(i)}, y_t), \quad (9)
$$

which implies

$$
\sum_{j=1}^{t-1} f(x^{(\ell)}, y_j) + \left\langle \Gamma^{(\ell)}, \alpha_t \right\rangle + \left\langle \Gamma^{(\ell)}, s^{(i)} \right\rangle - \left( \sum_{j=1}^{t-1} f(x^{(i)}, y_j) + \left\langle \Gamma^{(i)}, \alpha_t \right\rangle + \left\langle \Gamma^{(i)}, s^{(i)} \right\rangle \right)
$$

$$
\overset{(1)}{\leq} (f(x^{(i)}, y_t) - f(x^{(\ell)}, y_t)) + \left( \left\langle \Gamma^{(\ell)}, s^{(i)} \right\rangle - \left\langle \Gamma^{(i)}, s^{(i)} \right\rangle \right) \overset{(2)}{\leq} 0.
$$

(10)

Above, the first inequality comes from (9) and the second inequality is based on Definition 2. This completes the proof of Lemma 9. □

Combining (6) and Lemma 9, we get

$$
\mathbb{P}[x_t = x^{(i)}]
$$

$$
\leq \exp\left( \eta_t \|s^{(i)}\|_1 \right) \int_{\alpha_t} \mathbb{I}\left[\left\{ \operatorname*{argmin}_{k \in [K]} \sum_{j=1}^{t-1} f(x^{(k)}, y_j) + \left\langle \Gamma^{(k)}, \alpha_t + s^{(i)} \right\rangle \right\} = x^{(i)}\right] p(\alpha_t)\, d\alpha_t
$$

$$
\leq \exp\left( \eta_t \|s^{(i)}\|_1 \right) \int_{\alpha_t} \mathbb{I}\left[\left\{ \operatorname*{argmin}_{k \in [K]} \sum_{j=1}^{t} f(x^{(k)}, y_j) + \left\langle \Gamma^{(k)}, \alpha_t \right\rangle \right\} = x^{(i)}\right] p(\alpha_t)\, d\alpha_t
$$

$$
= \exp\left( \eta_t \|s^{(i)}\|_1 \right) \mathbb{P}[x'_t = x^{(i)}] \leq \exp(\gamma \eta_t) \mathbb{P}[x'_t = x^{(i)}].
$$

This completes the proof. □

### A.3.2 Proof of Lemma 7

We now use Lemma 2 to prove Lemma 7. Lemma 2 gives us

$$\mathbb{E}[f(x_t, y_t)] \leq \exp\left(\gamma\eta_t\right)\mathbb{E}[f(x'_t, y_t)] \leq \mathbb{E}[f(x'_t, y_t)] + 2\gamma\eta_t\mathbb{E}[f(x'_t, y_t)], \tag{11}$$

Above, the second inequality uses the fact that $\gamma\eta_t \leq 1$ and $\exp(x) \leq 1 + 2x$ for any $x \in [0, 1]$. Next, we focus on bounding the second term in the R.H.S. of (11). We have

$$2\gamma \sum_{t=1}^{T} \eta_t \mathbb{E}[f(x'_t, y_t)]$$

$$\stackrel{(1)}{\leq} 2\gamma \sum_{t=1}^{T} \eta_t \mathbb{E}\left[ f(x'_t, y_t) + \left(\sum_{j=1}^{t-1} f(x'_t, y_j) + \left\langle \Gamma^{(x'_t)}, \alpha_t \right\rangle\right) - \left(\sum_{j=1}^{t-1} f(x_t, y_j) + \left\langle \Gamma^{(x_t)}, \alpha_t \right\rangle\right) \right]$$

$$= 2\gamma \sum_{t=1}^{T} \eta_t \mathbb{E}\left[ \left(\sum_{j=1}^{t} f(x'_t, y_j) + \left\langle \Gamma^{(x'_t)}, \alpha_t \right\rangle\right) - \left(\sum_{j=1}^{t-1} f(x_t, y_j) + \left\langle \Gamma^{(x_t)}, \alpha_t \right\rangle\right) \right]$$

$$\stackrel{(2)}{\leq} 2\gamma \sum_{t=1}^{T} \eta_t \mathbb{E}\left[ \left(\sum_{j=1}^{t} f(x_{t+1}, y_j) + \left\langle \Gamma^{(x_{t+1})}, \alpha_t \right\rangle\right) - \left(\sum_{j=1}^{t-1} f(x_t, y_j) + \left\langle \Gamma^{(x_t)}, \alpha_t \right\rangle\right) \right]$$

$$= 2\gamma \sum_{t=1}^{T} \eta_t \mathbb{E}\left[ \left(\sum_{j=1}^{t} f(x_{t+1}, y_j) + \left\langle \Gamma^{(x_{t+1})}, \alpha_{t+1} \right\rangle\right) - \left(\sum_{j=1}^{t-1} f(x_t, y_j) + \left\langle \Gamma^{(x_t)}, \alpha_t \right\rangle\right) \right]$$

$$+ 2\gamma \sum_{t=1}^{T} \eta_t \left(\frac{1}{\eta_t} - \frac{1}{\eta_{t+1}}\right) \mathbb{E}\left[ \Gamma^{(x_{t+1})}\alpha \right]$$

$$\stackrel{(3)}{\leq} 2\gamma \sum_{t=1}^{T} \eta_t \mathbb{E}\left[ \left(\sum_{j=1}^{t} f(x_{t+1}, y_j) + \left\langle \Gamma^{(x_{t+1})}, \alpha_{t+1} \right\rangle\right) - \left(\sum_{j=1}^{t-1} f(x_t, y_j) + \left\langle \Gamma^{(x_t)}, \alpha_t \right\rangle\right) \right]$$

$$+ 2\gamma \sum_{t=1}^{T} \eta_t \left(\frac{1}{\eta_t} - \frac{1}{\eta_{t+1}}\right) \mathbb{E}\left[ \min_{i\in[K]} \Gamma^{(i)}\alpha \right]$$

$$= 2\gamma\eta_T \cdot \mathbb{E}\left[ \sum_{j=1}^{T} f(x_{T+1}, y_j) + \left\langle \Gamma^{(x_{T+1})}, \alpha_{T+1} \right\rangle \right] + 2\gamma \sum_{t=1}^{T} \eta_t \left(\frac{1}{\eta_t} - \frac{1}{\eta_{t+1}}\right) \cdot \mathbb{E}\left[ \min_{i\in[K]} \Gamma^{(i)}\alpha \right]$$

$$+ 2\gamma \sum_{t=1}^{T-1} (\eta_{t-1} - \eta_t) \cdot \mathbb{E}\left[ \sum_{j=1}^{t-1} f(x_t, y_j) + \left\langle \Gamma^{(x_t)}, \alpha_t \right\rangle \right] - 2\gamma\eta_1 \cdot \mathbb{E}\left[ \min_{i\in[K]} \Gamma^{(i)}\alpha \right]$$

$$\stackrel{(4)}{\leq} 2\gamma\eta_T \cdot \mathbb{E}\left[ \sum_{j=1}^{T} f(x^*, y_j) + \left\langle \Gamma^{(x^*)}, \alpha_{T+1} \right\rangle \right] + 2\gamma \sum_{t=1}^{T} \eta_t \left(\frac{1}{\eta_t} - \frac{1}{\eta_{t+1}}\right) \cdot \mathbb{E}\left[ \min_{i\in[K]} \Gamma^{(i)}\alpha \right]$$

$$+ 2\gamma \sum_{t=1}^{T-1} (\eta_{t-1} - \eta_t) \cdot \mathbb{E}\left[ \sum_{j=1}^{t-1} f(x^*, y_j) + \left\langle \Gamma^{(x^*)}, \alpha_t \right\rangle \right] - 2\gamma\eta_1 \cdot \mathbb{E}\left[ \min_{i\in[K]} \Gamma^{(i)}\alpha \right]$$

$$\stackrel{(5)}{\leq} 2\gamma\eta_T L_T^* + 2\gamma \sum_{t=1}^{T-1} (\eta_{t-1} - \eta_t) L_{t-1}^* + 2\gamma \sum_{t=1}^{T} \eta_t \left(\frac{1}{\eta_t} - \frac{1}{\eta_{t+1}}\right) \cdot \mathbb{E}\left[ \min_{i\in[K]} \Gamma^{(i)}\alpha \right]$$

$$- 2\gamma\eta_1 \cdot \mathbb{E}\left[ \min_{i\in[K]} \Gamma^{(i)}\alpha \right]. \tag{12}$$

Above, inequality (1) is based on the optimality of $x_t$, inequality (2) is due to the optimality of $x'_t$, inequality (3) is because $\frac{1}{\eta_t} - \frac{1}{\eta_{t+1}} \leq 0$ and $\mathbb{E}[\Gamma^{(x_{t+1})}\alpha] \geq \mathbb{E}[\min_{i\in[K]} \Gamma^{(i)}\alpha]$, inequality (4) is

based on the optimality of $x_t$, and the final inequality (5) is due to the fact that $x^*$ is independent of $\alpha$ and $\alpha$ is zero-mean. Next, we bound each term in the R.H.S. of the above equation respectively.

For the second term, denote $z_t = \max\{\gamma, \frac{1}{c}\sqrt{L_{t-1}^* + 1}\} = \frac{1}{\eta_t}$. WLOG, we assume the upper bound in Lemma 2 holds for a large enough $\gamma$ such that $\gamma \geq 1$. Then, we have

$$
\begin{aligned}
2\gamma \sum_{t=1}^{T-1}(\eta_{t-1} - \eta_t)L_{t-1}^* &\overset{(1)}{\leq} 2\gamma \sum_{t=1}^{T-1}\left(\frac{1}{z_{t-1}} - \frac{1}{z_t}\right)z_t^2 \\
&= 2\gamma \sum_{t=1}^{T-1}\frac{(z_t - z_{t-1})z_t^2}{z_t z_{t-1}} \\
&= 2\gamma \sum_{t=1}^{T-1}\frac{(z_t^2 - z_{t-1}^2)z_t}{z_{t-1}(z_t + z_{t-1})} \\
&= 2\gamma \sum_{t=1}^{T-1}\frac{(z_t^2 - z_{t-1}^2)\left((z_t - z_{t-1}) + z_{t-1}\right)}{z_{t-1}(z_t + z_{t-1})} \\
&= 2\gamma \sum_{t=1}^{T-1}\left(\frac{(z_t^2 - z_{t-1}^2)^2}{z_{t-1}(z_t + z_{t-1})^2} + z_t - z_{t-1}\right) \\
&\overset{(2)}{\leq} 2\gamma \sum_{t=1}^{T-1}\left(\frac{(z_t^2 - z_{t-1}^2)}{z_{t-1}^2} + z_t - z_{t-1}\right) \\
&\overset{(3)}{\leq} 2\gamma \sum_{t=1}^{T-1}\left(4(\ln(z_t) - \ln(z_{t-1})) + (z_t - z_{t-1})\right) \\
&= 8\gamma \cdot \ln\left(\frac{z_{T-1}}{z_0}\right) + 2\gamma(z_{T-1} - z_0) \\
&\overset{(4)}{\leq} 8\gamma \cdot \ln\left(\frac{1}{c}\sqrt{L_T^* + 1} + \gamma\right) + 2\gamma\left(\frac{1}{c}\sqrt{L_T^* + 1} + \gamma\right).
\end{aligned}
\tag{13}
$$

Above, inequality (2) is due to the fact that $z_t + z_{t-1} \geq 1$, $0 \leq z_t^2 - z_{t-1}^2 \leq 1$ and $z_t \geq z_{t-1}$, inequality (3) is based on the identity $x \leq 2\ln(1 + x)$ for $x \leq 1$, and the last inequality (4) follows from the definition of $z_t$.

We now control the last two terms of (12). Since the distribution of $\alpha$ is symmetric, the distributions of $\alpha$ and $-\alpha$ are the same. Thus, we have

$$
\mathbb{E}\left[\min_{i \in [K]} \Gamma^{(i)}\alpha\right] = \mathbb{E}\left[\min_{i \in [K]} -\Gamma^{(i)}\alpha\right] = -\mathbb{E}\left[\max_{i \in [K]} \Gamma^{(i)}\alpha\right],
$$

which gives us

$$
\begin{aligned}
&2\gamma \sum_{t=1}^{T}\eta_t\left(\frac{1}{\eta_t} - \frac{1}{\eta_{t+1}}\right) \cdot \mathbb{E}\left[\min_{i \in [K]} \Gamma^{(i)}\alpha\right] - 2\gamma\eta_1 \mathbb{E}\left[\min_{i \in [K]} \Gamma^{(i)}\alpha\right] \\
&\leq 2\gamma \max\left\{\mathbb{E}\left[\max_{i \in [K]} \Gamma^{(i)}\alpha\right], 0\right\}\left(\frac{\eta_1}{\eta_{T+1}} - 1 + 1\right) \\
&\leq \frac{2}{\eta_{T+1}} \max\left\{\mathbb{E}\left[\max_{i \in [K]} \Gamma^{(i)}\alpha\right], 0\right\}.
\end{aligned}
\tag{14}
$$

Finally, we leverage the following lemma to complete the proof.

**Lemma 10.** *We have*

$$
\mathbb{E}\left[\max_{i \in [K]} \Gamma^{(i)}\alpha\right] \leq \sqrt{2}\max\{2\ln K, \sqrt{N\ln K}\}.
$$

A direct substitution of Lemma 10 obtains the desired control on the last two terms of (12) and completes the proof. It only remains to prove Lemma 10 which we do below.

*Proof.* Let $\beta^{(k)} = \sum_{i=1}^{N} \Gamma^{(k,i)} \alpha^{(i)}$, and we have for $\lambda < 1$,

$$
\begin{aligned}
\mathbb{E}\left[\max_{k \in [K]} \Gamma^{(k)} \alpha\right] = \mathbb{E}\left[\max_{k \in [K]} \beta^{(k)}\right] &= \frac{1}{\lambda} \ln\left(\exp\left(\lambda \mathbb{E}\left[\max_{k \in [K]} \beta^{(k)}\right]\right)\right) \\
&\overset{(1)}{\leq} \frac{1}{\lambda} \ln\left(\mathbb{E}\left[\exp\left(\lambda \max_{k \in [K]} \beta^{(k)}\right)\right]\right) \\
&\overset{(2)}{\leq} \frac{1}{\lambda} \ln\left(\sum_{k \in [K]} \mathbb{E}\left[\exp\left(\lambda \beta^{(k)}\right)\right]\right) \\
&= \frac{1}{\lambda} \ln\left(\sum_{k \in [K]} \mathbb{E}\left[\exp\left(\sum_{i=1}^{N} \lambda \Gamma^{(k,i)} \alpha^{(i)}\right)\right]\right) \\
&\overset{(3)}{\leq} \frac{1}{\lambda} \ln\left(K\left(\frac{1}{1-\lambda^2}\right)^N\right) \\
&= \frac{\ln K}{\lambda} + \frac{N}{\lambda} \ln\left(\frac{1}{1-\lambda^2}\right) \\
&\overset{(4)}{\leq} \min_{\lambda \in \left(0, \frac{\sqrt{2}}{2}\right]}\left[\frac{\ln K}{\lambda} + 2\lambda N\right] \leq \sqrt{2} \max\{2\ln K, \sqrt{N \ln K}\}.
\end{aligned}
$$
(15)

Above, inequality $(1)$ is based on Jensen's inequality, inequality $(3)$ follows from the expression of the moment-generating function of a Laplace distribution, and the final inequality $(4)$ follows from the identity $\ln(1/(1-x)) \leq 2x$ for $x \in \left(0, \frac{1}{2}\right]$. This completes the proof. $\square$

### A.3.3 Proof of Lemma 8

Recall that the infeasible leader is given by

$$
\begin{aligned}
x'_t &= \operatorname*{argmin}_{k \in [K]} \sum_{j=1}^{t} f(x^{(k)}, y_j) + \left\langle \Gamma^{(k)}, \alpha_t \right\rangle \\
&= \operatorname*{argmin}_{k \in [K]} \sum_{j=1}^{t} \left( f(x^{(k)}, y_j) + \left\langle \Gamma^{(k)}, \alpha_j \right\rangle - \left\langle \Gamma^{(k)}, \alpha_{j-1} \right\rangle \right).
\end{aligned}
$$
(16)

Recall that we set $\alpha_0 = 0$. Then, we have

$$
\begin{aligned}
\sum_{t=1}^{T} f(x'_t, y_t) + \Gamma^{(x'_t)} \alpha_t - \Gamma^{(x'_t)} \alpha_{t-1} &\overset{(1)}{\leq} \min_{k \in [K]} \sum_{t=1}^{T} \left( f(x^{(k)}, y_t) + \Gamma^{(k)} \alpha_t - \Gamma^{(k)} \alpha_{t-1} \right) \\
&= \min_{k \in [K]} \left( \sum_{t=1}^{T} f(x^{(k)}, y_t) + \Gamma^{(k)} \alpha_T \right) \\
&\overset{(2)}{\leq} \sum_{t=1}^{T} f(x^*, y_t) + \Gamma^* \alpha_T \\
&\overset{(3)}{\leq} \sum_{t=1}^{T} f(x^*, y_t) + \max_{k \in [K]} \Gamma^{(k)} \alpha_T,
\end{aligned}
$$
(17)

where inequaliy $(1)$ is based on Lemma 3.1 of (Cesa-Bianchi & Lugosi, 2006). Because the learning rate sequence $\{\eta_t\}_{t \geq 1}$ is non-increasing, we have $\alpha_{t-1} - \alpha_t \geq 0$. Thus, we get

$$
\sum_{t=1}^{T} f(x'_t, y_t) - f(x^*, y_t) \leq \max_{k \in [K]} \Gamma^{(k)} \alpha_T + \sum_{t=1}^{T} \max_{k \in [K]} \Gamma^{(k)} \alpha \cdot \left(\frac{1}{\eta_{t-1}} - \frac{1}{\eta_t}\right).
$$
(18)

Taking an expectation on both sides with respect to the randomness in the algorithm yields

$$\mathbb{E}\left[\sum_{t=1}^{T} f(x'_t, y_t) - f(x^*, y_t)\right] \leq \mathbb{E}\left[\max_{k\in[K]} \Gamma^{(k)}\alpha_T\right] + \mathbb{E}\left[\sum_{t=1}^{T} \max_{k\in[K]} \Gamma^{(k)}\alpha \cdot \left(\frac{1}{\eta_{t-1}} - \frac{1}{\eta_t}\right)\right]$$

$$= \mathbb{E}\left[\max_{k\in[K]} \Gamma^{(k)}\alpha_T\right] + \sum_{t=1}^{T}\left(\frac{1}{\eta_t} - \frac{1}{\eta_{t-1}}\right) \cdot \mathbb{E}\left[\max_{k\in[K]} \Gamma^{(k)}\alpha\right] \quad (19)$$

$$\leq 2\frac{\max\{\mathbb{E}[\max_{i\in[K]} \Gamma^{(i)}\alpha], 0\}}{\eta_T},$$

where the first equality follows because the distribution of the Laplace noise is symmetric. The proof is finished by combining the above inequality with Lemma 10. □

### A.4 Lower Bound for GFTPL

In this part, we introduce the lower bound for GFTPL. We first prove the following lemma.

**Lemma 11.** *Denote*

$$x_t^* = \underset{k\in[K]}{\operatorname{argmin}} \sum_{s=1}^{t} f(x^{(k)}, y_s), \quad (20)$$

*then we have*

$$\Gamma^{(x_{t+1})}\alpha \leq \Gamma^{(x_t^*)}\alpha. \quad (21)$$

*Proof.* Considering the definitions of $x_{t+1}$ and $x_t^*$, we have:

$$\sum_{s=1}^{t} f(x_{t+1}, y_s) \geq \sum_{s=1}^{t} f(x_t^*, y_s), \quad (22)$$

and

$$\sum_{s=1}^{t} f(x_t^*, y_s) + \Gamma^{(x_t^*)}\frac{\alpha}{\eta_{t+1}} \geq \sum_{s=1}^{t} f(x_{t+1}, y_s) + \Gamma^{(x_{t+1})}\frac{\alpha}{\eta_{t+1}}. \quad (23)$$

The required inequality can be shown by adding up the above two inequalities. □

We prove the lower bound result as follows.

**Theorem 4.** *Assume $\Gamma$ is $\gamma$-approximable, then Algorithm 1 with $\eta_t = \min\left\{\frac{1}{\gamma}, \frac{c}{\sqrt{L_{t-1}^*+1}}\right\}$, where $L_{t-1}^* = \min_{k\in[K]} \sum_{j=1}^{t-1} f(x^{(k)}, y_j)$ has the following regret lower bound:*

$$R_T = \mathbb{E}\left[\sum_{t=1}^{T} f(x_t, y_t) - \sum_{t=1}^{T} f(x^*, y_t)\right] \geq -2\sqrt{2}\max\{2\ln K, \sqrt{N\ln K}\}\left(\gamma + \frac{1}{c}\sqrt{L_T^*+1}\right).$$

*Proof.* We first show an intermediate conclusion via induction:

$$\sum_{t=1}^{T} f(x_t, y_t) + \Gamma^{(x_t)}\alpha\left(\frac{1}{\eta_{t+1}} - \frac{1}{\eta_t}\right) \geq \sum_{t=1}^{T} f(x_{T+1}, y_t) + \Gamma^{(x_{T+1})}\frac{\alpha}{\eta_{T+1}} - \Gamma^{(x_1)}\frac{\alpha}{\eta_1}, \quad (24)$$

which obviously holds for $T = 1$. Assume this holds for $T - 1$:

$$\sum_{t=1}^{T-1} f(x_t, y_t) + \Gamma^{(x_t)}\alpha\left(\frac{1}{\eta_{t+1}} - \frac{1}{\eta_t}\right) \geq \sum_{t=1}^{T-1} f(x_T, y_t) + \Gamma^{(x_T)}\frac{\alpha}{\eta_T} - \Gamma^{(x_1)}\frac{\alpha}{\eta_1}. \quad (25)$$

Noticing

$$\sum_{t=1}^{T} f(x_T, y_t) + \Gamma^{(x_T)}\frac{\alpha}{\eta_{T+1}} \geq \sum_{t=1}^{T} f(x_{T+1}, y_t) + \Gamma^{(x_{T+1})}\frac{\alpha}{\eta_{T+1}}, \quad (26)$$

by rearranging we can show

$$f(x_T, y_T) + \Gamma^{(x_T)}\alpha\left(\frac{1}{\eta_{T+1}} - \frac{1}{\eta_T}\right) \geq \sum_{t=1}^{T} f(x_{T+1}, y_t) + \Gamma^{(x_{T+1})}\frac{\alpha}{\eta_{T+1}}$$
$$- \left(\sum_{t=1}^{T-1} f(x_T, y_t) + \Gamma^{(x_T)}\frac{\alpha}{\eta_T}\right) \tag{27}$$

Adding up (25) and (27) we can prove the required conclusion for round $T$.

Combining (24) and

$$\sum_{t=1}^{T} f(x_{T+1}, y_t) \geq \sum_{t=1}^{T} f(x_T^*, y_t), \tag{28}$$

we have

$$\sum_{t=1}^{T} \left(f(x_t, y_t) - f(x_T^*, y_t)\right)$$
$$\geq \sum_{t=1}^{T} \left(f(x_t, y_t) - f(x_{T+1}, y_t)\right)$$
$$\geq \Gamma^{(x_{T+1})}\frac{\alpha}{\eta_{T+1}} - \Gamma^{(x_1)}\frac{\alpha}{\eta_1} - \sum_{t=1}^{T} \Gamma^{(x_t)}\alpha\left(\frac{1}{\eta_{t+1}} - \frac{1}{\eta_t}\right) \tag{29}$$
$$\geq \Gamma^{(x_{T+1})}\frac{\alpha}{\eta_{T+1}} - \Gamma^{(x_1)}\frac{\alpha}{\eta_1} - \sum_{t=1}^{T} \Gamma^{(x_{t-1}^*)}\alpha\left(\frac{1}{\eta_{t+1}} - \frac{1}{\eta_t}\right)$$
$$\geq -2\frac{\max_{i\in[K]} \Gamma^{(i)}\alpha}{\eta_{T+1}} - \sum_{t=1}^{T} \Gamma^{(x_{t-1}^*)}\alpha\left(\frac{1}{\eta_{t+1}} - \frac{1}{\eta_t}\right),$$

where for the third inequality, Lemma 11 is adopted while for the fourth, we use the symmetry of $\alpha$ and the non-increasing property of $\eta_t$. Now we can take the expectation and get

$$\mathbb{E}\left[\sum_{t=1}^{T} \left(f(x_t, y_t) - f(x_T^*, y_t)\right)\right] \geq -2\mathbb{E}\left[\frac{\max_{i\in[K]} \Gamma^{(i)}\alpha}{\eta_{T+1}}\right]$$
$$\geq -2\sqrt{2}\max\{2\ln K, \sqrt{N\ln K}\}\left(\gamma + \frac{1}{c}\sqrt{L_T^* + 1}\right), \tag{30}$$

where we use Lemma 10, $\mathbb{E}[\alpha] = 0$ and $\eta_t = \min\left\{\frac{1}{\gamma}, \frac{c}{\sqrt{L_{t-1}^* + 1}}\right\}$. $\qquad\square$

**Remark** Combining Theorems 1 and 4 while setting $c = 1$, we have

$$-O\left(\frac{\max\left\{\ln K, \sqrt{N\ln K}\right\}}{\sqrt{L_T^*}}\right) \leq \frac{\mathbb{E}\left[\sum_{t=1}^{T} f(x_t, y_t)\right]}{L_T^*} - 1 \leq O\left(\frac{\max\left\{\gamma, \ln K, \sqrt{N\ln K}\right\}}{\sqrt{L_T^*}}\right). \tag{31}$$

As $L_T^*$ goes to $\infty$, both sides go to 0, which means our strategy competes the best expert in hindsight.

### A.5 Extension to $\ell_p$ Perturbation

In this section, we extend our techniques to perturbation distributions that are exponential with respect to an $\ell_p$-norm for any $p \geq 1$ (note that $p = 1$ corresponds to the case of the Laplace

distribution). Specifically, we consider the probability density function

$$p(\alpha) \propto \exp\left\{ -\left( \sum_i |\alpha^{(i)}|^p \right)^{\frac{1}{p}} \right\}.$$ (32)

Recall that we have the following decomposition of regret:

$$
R_T = \mathbb{E}\left[ \sum_{t=1}^{T} f(x_t, y_t) - f(x^*, y_t) \right]
$$

$$
= \underbrace{\mathbb{E}\left[ \sum_{t=1}^{T} f(x_t, y_t) - \sum_{t=1}^{T} f(x'_t, y_t) \right]}_{\text{TERM 1}} + \underbrace{\mathbb{E}\left[ \sum_{t=1}^{T} f(x'_t, y_t) - \sum_{t=1}^{T} f(x^*, y_t) \right]}_{\text{TERM 2}},
$$ (33)

A critical observation is that the proof of Lemma 2 relies on the triangle inequality

$$\frac{p(\beta)}{p(\beta - s^{(i)})} = \exp\left( \eta_t \left( \left\| \beta - s^{(i)} \right\|_1 - \|\beta\|_1 \right) \right) \le \exp\left( \eta_t \|s^{(i)}\|_1 \right),$$

which is easily generalized to the $\ell_p$-norm:

$$\frac{p(\beta)}{p(\beta - s^{(i)})} = \exp\left( \eta_t \left( \left\| \beta - s^{(i)} \right\|_p - \|\beta\|_p \right) \right) \le \exp\left( \eta_t \|s^{(i)}\|_p \right).$$

Following the proof of Lemma 2, we then get

$$\mathbb{E}[f(x_t, y_t)] \le \exp\left( \gamma_p \eta_t \right) \mathbb{E}[f(x'_t, y_t)],$$ (34)

where $\gamma_p$ is now an upper bound on $\|s^{(i)}\|_p$. As before, noting that $\gamma_p \eta_t \le 1$ and $\exp(x) \le 1 + 2x$ for any $x \in [0, 1]$ gives us

$$\mathbb{E}[f(x_t, y_t)] \le \exp\left( \gamma_p \eta_t \right) \mathbb{E}[f(x'_t, y_t)] \le (1 + 2\gamma_p \eta_t) \mathbb{E}[f(x'_t, y_t)].$$ (35)

It remains to upper bound $\mathbb{E}\left[ \max_{i \in [K]} \Gamma^{(i)} \alpha \right]$ under the $\ell_p$ perturbation (as was previously done for the Laplace case). This is done in the following lemma.

**Lemma 12.** *Under the $\ell_p$ perturbation, we have*

$$\mathbb{E}\left[ \max_{i \in [K]} \Gamma^{(i)} \alpha \right] \le 2N^{1-\frac{1}{p}}(\ln K + N \ln 2).$$ (36)

*Proof.* Similar to the proof of Lemma 10, we have

$$\mathbb{E}\left[ \max_{i \in [K]} \Gamma^{(i)} \alpha \right] \le \frac{1}{\lambda} \ln\left( K \cdot \mathbb{E}\left[ \exp\left( \sum_{i=1}^{N} \lambda |\alpha^{(i)}| \right) \right] \right),$$ (37)

where we use the fact that $\Gamma \in [0, 1]^{K \times N}$. Now we calculate

$$
\mathbb{E}\left[ \exp\left( \sum_{i=1}^{N} \lambda |\alpha^{(i)}| \right) \right] = \frac{\int \exp(\lambda \|\alpha\|_1) \cdot \exp(-\|\alpha\|_p) d\alpha}{\int \exp(-\|\alpha\|_p)} d\alpha
$$

$$
\le \frac{\int \exp(-(1 - \lambda N^{1-\frac{1}{p}})\|\alpha\|_p) d\alpha}{\int \exp(-\|\alpha\|_p) d\alpha},
$$ (38)

where the norm inequality $\|\alpha\|_1 \le N^{1-\frac{1}{p}} \|\alpha\|_p$ is used. Setting $\lambda = \frac{1}{2N^{1-\frac{1}{p}}}$ gives us

$$\mathbb{E}\left[ \exp\left( \sum_{i=1}^{N} \lambda |\alpha^{(i)}| \right) \right] \le 2^N,$$ (39)

and thus

$$\mathbb{E}\left[ \max_{i \in [K]} \Gamma^{(i)} \alpha \right] \le 2N^{1-\frac{1}{p}}(\ln K + N \ln 2).$$ (40)

$\square$

We now complete the proof extension. According to the proof of Lemma 5, for TERM 1 we have

$$\mathbb{E}\left[\sum_{t=1}^{T} f(x_t, y_t) - \sum_{t=1}^{T} f(x'_t, y_t)\right] \leq 2\gamma_p \sum_{t=1}^{T} \eta_t \mathbb{E}[f(x'_t, y_t)]$$

$$\leq 2\gamma_p \left(\eta_T L_T^* + \sum_{t=1}^{T-1}(\eta_{t-1} - \eta_t)L_{t-1}^* + \sum_{t=1}^{T} \eta_t \left(\frac{1}{\eta_t} - \frac{1}{\eta_{t+1}}\right) \mathbb{E}\left[\min_{i \in [K]} \Gamma^{(i)}\alpha\right] - \eta_1 \mathbb{E}\left[\min_{i \in [K]} \Gamma^{(i)}\alpha\right]\right)$$

$$\leq 2\gamma_p \left(cN^{1-\frac{1}{p}}\sqrt{L_T^* + 1} + \left(\frac{1}{cN^{1-\frac{1}{p}}}\sqrt{L_T^* + 1} + \gamma_p\right) + 4\ln\left(\frac{1}{cN^{1-\frac{1}{p}}}\sqrt{L_T^* + 1} + \gamma_p\right)\right)$$

$$+2\phi\left(\gamma_p + \frac{\sqrt{L_T^* + 1}}{cN^{1-\frac{1}{p}}}\right),$$

(41)

where $\phi$ denotes an upper bound on $\mathbb{E}\left[\max_{i \in [K]} \Gamma^{(i)}\alpha\right]$ that will be specified shortly. Above, we plug in $\eta_t = \min\left\{\frac{1}{\gamma_p}, \frac{cN^{1-\frac{1}{p}}}{\sqrt{L_{t-1}^* + 1}}\right\}$ to get the third inequality. For TERM 2, a similar argument to the proof of Lemma 8 gives

$$\mathbb{E}\left[\sum_{t=1}^{T} f(x'_t, y_t) - \sum_{t=1}^{T} f(x^*, y_t)\right] \leq 2\phi\left(\gamma_p + \frac{\sqrt{L_T^* + 1}}{cN^{1-\frac{1}{p}}}\right). \tag{42}$$

Thus, the total regret is upper bounded by

$$R_T \leq 2\gamma_p \left(cN^{1-\frac{1}{p}}\sqrt{L_T^* + 1} + \left(\frac{1}{cN^{1-\frac{1}{p}}}\sqrt{L_T^* + 1} + \gamma_p\right) + 4\ln\left(\frac{1}{cN^{1-\frac{1}{p}}}\sqrt{L_T^* + 1} + \gamma_p\right)\right)$$

$$+4\phi\left(\gamma_p + \frac{\sqrt{L_T^* + 1}}{cN^{1-\frac{1}{p}}}\right) = O\left(2\gamma_p\left(cN^{1-\frac{1}{p}} + \frac{1}{cN^{1-\frac{1}{p}}}\right)\sqrt{L_T^* + 1} + 4\phi\frac{\sqrt{L_T^* + 1}}{cN^{1-\frac{1}{p}}}\right).$$

(43)

If we use a $\ell_p$ perturbation, by Lemma 12, we have $\phi = 2N^{1-\frac{1}{p}}(\ln K + N \ln 2)$ and

$$R_T = O\left(2\gamma_p\left(cN^{1-\frac{1}{p}} + \frac{1}{cN^{1-\frac{1}{p}}}\right)\sqrt{L_T^* + 1} + 8N^{1-\frac{1}{p}}(\ln K + N \ln 2)\frac{\sqrt{L_T^* + 1}}{cN^{1-\frac{1}{p}}}\right)$$

$$= O\left(\max\left\{\gamma_p N^{1-\frac{1}{p}}, \ln K, N\right\}\sqrt{L_T^*}\right).$$

(44)

which completes the proof. □

We do a brief comparison between the $\ell_p$-perturbation and Laplace perturbation for the case when $\Gamma \in \{0, 1\}^{K \times N}$ is a binary matrix. By Lemma 3 $\gamma_p = N^{\frac{1}{p}}$ because $s^{(i)} \in \{-1, 1\}^N$. Then we get that the regret under the $\ell_p$ perturbation is

$$R_T = O\left(\max\{N, \ln K\}\sqrt{L_T^*}\right),$$

while by Theorem 1 the regret bound under the Laplace distribution is

$$R_T = O\left(\max\left\{N, \ln K, \sqrt{N \ln K}\right\}\sqrt{L_T^*}\right).$$

We can see the regret bounds are the same. Since $\ell_p$ perturbation does not lead to an improvement on the regret bound and the Laplace distribution is easier to sample, we only consider the Laplace distribution in the main paper.

## B    Omitted Proof for Section 4

In this section, we provide the omitted proofs for Section 4.

## B.1 Proof of Lemma 3

We begin by proving Lemma 3, which shows that any $\{0,1\}$-valued PTM with distinct rows satisfies $\gamma$-approximability. We first state the following lemma which introduces a slightly stronger condition for $\gamma$-approximability.

**Lemma 13.** *Let $\Gamma \in [0,1]^{K \times N}$ be a matrix, and denote $\Gamma^{(k)}$ as the k-th row of $\Gamma$. If $\forall k \in [K]$, $\exists s \in \mathbb{R}^N$, $\|s\|_1 \leq \gamma$, such that $\langle \Gamma^{(k)}, s \rangle - \langle \Gamma^{(j)}, s \rangle \geq 1$ for all rows $j \neq k$, then $\Gamma$ is $\gamma$-approximable.*

*Proof.* Since $\forall y \in \mathcal{Y}, k, j \in [K]$, $1 \geq f(x^{(k)}, y) - f(x^{(j)}, y)$, it is straightforward to see that the condition in Lemma 13 is a sufficient condition of Definition 2. $\square$

Next, we construct a $\gamma$-approximable $\Gamma$ based on Lemma 13. Denote $\Gamma^{(k,i)}$ as the $i$-th element of $\Gamma^{(k)}$. $\forall t > 0, k \in [K]$, we set $s^{(k)} = 2\Gamma^{(k)} - 1$. Since $\forall k \in [N]$, $s^{(k)} \in \{-1, 1\}^N$, we have $\|s^{(k)}\|_1 \leq N$. On the other hand, $\forall j \neq k$,

$$\left\langle \Gamma^{(k)}, s^{(k)} \right\rangle - \left\langle \Gamma^{(j)}, s^{(k)} \right\rangle = \sum_{i=1}^{N} (\Gamma^{(k,i)} - \Gamma^{(j,i)}) \cdot (2\Gamma^{(k,i)} - 1). \tag{45}$$

For each term $i$ in the R.H.S. of the equality, we have

$$(\Gamma^{(k,i)} - \Gamma^{(j,i)}) \cdot (2\Gamma^{(k,i)} - 1) = \begin{cases} 0, & \Gamma^{(k,i)} = \Gamma^{(j,i)}, \\ 1, & \Gamma^{(k,i)} \neq \Gamma^{(j,i)}. \end{cases}$$

Note that since every two rows of $\Gamma$ differ by at least one element, there must exist one $i \in [N]$ such that $(\Gamma^{(k,i)} - \Gamma^{(j,i)}) \cdot (2\Gamma^{(k,i)} - 1) = 1$. This completes the proof of the lemma. $\square$

Lemma 3 is simple but powerful, and can be applied to a broad variety of combinatorial auction problems. This is detailed next.

## B.2 Auction Problems with a Binary $\Gamma$

Imagine that a seller wants to sell $k$ items (that are either homogeneous or heterogeneous) to $n$ bidders. Each bidder has a combinatorial utility function $b^{(i)} : \{0,1\}^k \to [0,1]$ and we use $b$ to denote the bidding profile vector of all bidders. In this work we consider *truthful* auctions, i.e. each bidder is incentivized to report his true valuation $b^{(i)}$ in the unique Bayes-Nash equilibrium of the auction. The $i$-th bidder gets an allocation $q^{(i)}(b) \in \{0,1\}^k$ and pays the seller $p^{(i)}(b)$. Therefore, the utility of the bidder is given by $b^{(i)}(q^{(i)}(b)) - p^{(i)}(b)$.

An auction $a$ receives the bidding profiles of all bidders and determines how to allocate the items and how much to charge each bidder. We use $r(a, b) := \sum_{i=1}^{n} p^{(i)}(b)$ to denote the revenue yielded by applying auction $a$ to the bidder profile $b$. We consider a *repeated auction* setting in which the auctioneer faces different bidders on each round. The bidders may be of very heterogeneous types, so we do not make any assumptions on the bidder profile and assume that it can arbitrarily change from round to round. More formally: for each round $t = 1, \ldots, T$, the learner chooses an auction $a_t$ while the adversary chooses a bidder profile $b_t$. Then, the learner gets to know $b_t$ and receives the revenue $r(a_t, b_t)$. The goal of the learner is to compete the revenue earned by the best auction in hindsight. Following Dudík et al. (2020), if the revenue $r(a, b) \in [0, R]$ where $R > 1$, then we can scale all rewards by $\frac{1}{R}$ to ensure all rewards are in $[0, 1]$. After applying Algorithm 2, we scale the reward back to get the $O(R\sqrt{T - L_T^*})$ regret.

Now we briefly introduce auction problems that admit a binary-valued TPM $\Gamma$. By Lemma 1 these are $\gamma$-approximable and by Theorem 1 these admit small-loss bounds.

**VCG with bidder-specific reserves** For the standard VCG auction, multiple bidders can be simultaneously served if the allocation $q_*$ maximizes the total social welfare $\sum_{i=1}^{n} b^{(i)} q_*^{(i)}$. Then the bidder who wins a set of items would pay the externality he imposes on others

$$p^{(i)}(b) = \max_q \left( \sum_{j \neq i} b^{(j)} q^{(j)} \right) - \sum_{j \neq i} b^{(j)} q_*^{(j)}.$$

The setting we discuss is slightly modified in the sense that we have a vector $a$ with $i$-th component being the reserve value of the $i$-th bidder. Any bidder whose valuation $b^{(i)}$ is smaller than $a^{(i)}$ will be eliminated. Then, we run the VCG auction for the remaining bidders.

Following Dudík et al. (2020), we discretize reserve prices and use the same $\Gamma$ therein to get the following small-loss bound:

**Theorem 5.** *We consider VCG auction with reserves for the single-item $s$-unit setting, and the set of all feasible auctions is denoted by $\mathcal{I}$. Denote $R = \max_{a,b} r(a, b)$. Let $\Gamma$ be an $|\mathcal{I}_m| \times n\lceil\log m\rceil$ binary matrix, where $\mathcal{I}_m$ contains auctions in which each reservation price comes from $\left\{\frac{1}{m}, \ldots, \frac{m}{m}\right\}$, and consecutive $\lceil\log m\rceil$ columns correspond to binary encodings of each bidder, then $\Gamma$ is implementable. Running Algorithm 2 with such a $\Gamma$ yields*

$$\mathbb{E}\left[\max_{a\in\mathcal{I}}\sum_{t=1}^{T} r(a, b_t) - \sum_{t=1}^{T} r(a_t, b_t)\right] = O\left(nR\sqrt{T - L_T^*}\log(Ts)\right). \tag{46}$$

*Proof.* The implementability of $\Gamma$ follows from Lemma 3.3 of Dudík et al. (2020). Since $\Gamma$ is binary and every two rows are distinct, by Lemma 3 we know it is $N$-approximable. Using Corollary 1 we have

$$\begin{aligned}
\mathbb{E}\left[\max_{a\in\mathcal{I}_m}\sum_{t=1}^{T} r(a, b_t) - \sum_{t=1}^{T} r(a_t, b_t)\right] &= O\left(R \cdot \max\{\gamma, \ln K, \sqrt{N \ln K}\}\sqrt{T - L_T^*}\right) \\
&= O\left(R \cdot \max\{N, \ln K, \sqrt{N \ln K}\}\sqrt{T - L_T^*}\right) \\
&= O\left(R \cdot N\sqrt{T - L_T^*}\right) \\
&= O\left(R \cdot n\log m\sqrt{T - L_T^*}\right),
\end{aligned} \tag{47}$$

where we use the facts that $N = n\lceil\log m\rceil$, $K = |\mathcal{I}_m|$, and $N = \Omega(\log K)$ since $\Gamma$ is binary. According to Dudík et al. (2020), the optimal revenue in $\mathcal{I}$ is upper bounded by that of $\mathcal{I}_m$:

$$\mathbb{E}\left[\max_{a\in\mathcal{I}}\sum_{t=1}^{T} r(a, b_t) - \max_{a\in\mathcal{I}_m}\sum_{t=1}^{T} r(a, b_t)\right] \leq \frac{Ts}{m}. \tag{48}$$

Combining (47) and (48) while setting $m = O(Ts)$ yield the proposed Theorem. $\square$

**Envy-free item pricing**  Assume there are $k$ different items and we use $a$ to denote the vector of each item's price. Bidders come one by one. The $i$-th bidder greedily chooses a bundle $q^{(i)} \in \{0, 1\}^k$ which maximizes his utility $b^{(i)}(q^{(i)}) - a \cdot q^{(i)}$ and pays $a \cdot q^{(i)}$. Similar as the VCG with bidder-specific reserves, we also assume each price is discretized in the set $a^{(i)} \in \{\frac{1}{m}, \ldots, \frac{m}{m}\}$.

**Theorem 6.** *We consider envy-free auction for $n$ single-minded bidders and $k$ heterogeneous items with infinite supply. Denote $\mathcal{P}$ to be the set of all possible auctions and $R = \max_{a,b} r(a, b)$. Let $\Gamma$ be an $|\mathcal{P}_m| \times (k\lceil\log m\rceil$ binary matrix, where $\mathcal{P}_m$ contains envy-free item auctions in which all prices come from $\left\{\frac{1}{m}, \ldots, \frac{m}{m}\right\}$ and consecutive $\lceil\log m\rceil$ columns correspond to binary encodings of each item's price. Then, $\Gamma$ is implementable and running Algorithm 2 with this value of $\Gamma$ yields*

$$\mathbb{E}\left[\max_{a\in\mathcal{P}}\sum_{t=1}^{T} r(a, b_t) - \sum_{t=1}^{T} r(a_t, b_t)\right] = O\left(kR\sqrt{T - L_T^*}\log(kT)\right). \tag{49}$$

*Proof.* As noticed in Dudík et al. (2020), we can consider a bidder who has valuation $b^{(i)}$ for the bundle of the $i$-th item and valuations 0 for any other bundles. The revenue of auction $a$ on such a bidder profile is $a^{(i)}\mathbb{I}[b^{(i)} \geq a^{(i)}]$. Similarly, for the VCG auction with reserves $a$, the revenue of a bidder who has a non-zero valuation $b^{(i)}$ would be $a^{(i)}\mathbb{I}[b^{(i)} \geq a^{(i)}]$. Based on the equivalence between envy-free auction and VCG auction with reserves, we can apply Theorem 5 with $n = k$ to get the following bound.

$$\mathbb{E}\left[\max_{a\in\mathcal{P}_m}\sum_{t=1}^{T}r(a,b_t)-\sum_{t=1}^{T}r(a_t,b_t)\right]=O\left(R\cdot k\log m\sqrt{T-L_T^*}\right), \tag{50}$$

As also pointed out by Dudík et al. (2020), the optimal revenue of $\mathcal{P}$ would not be much larger than that of $\mathcal{P}_m$ upto a small discretization-related error:

$$\mathbb{E}\left[\max_{a\in\mathcal{P}}\sum_{t=1}^{T}r(a,b_t)-\max_{a\in\mathcal{P}_m}\sum_{t=1}^{T}r(a,b_t)\right]\leq\frac{nk^2T}{m}. \tag{51}$$

Combining Equations 50 and 51 while setting $m=O(k^2T)$ yields the theorem. $\qquad\square$

**Online welfare maximization for multi-unit items**  In this setting, we wish to allocate $h\gg n$ homogeneous items to $n$ bidders such that $\sum_{i=1}^{n}a^{(i)}=h$. Each bidder has a valuation function $b^{(i)}:\mathbb{N}\to[0,1]$ that maps the number of items he obtains to the utility. We assume $b^{(i)}$ is non-decreasing and $b^{(i)}(0)=0$. The objective is to maximize the total social welfare $\sum_{i=1}^{n}b^{(i)}(a^{(i)})$.

We denote the set of allocations which satisfy $\sum_{i=1}^{n}a^{(i)}=s$ as $\mathcal{X}$. For the offline version of this problem, Dobzinski & Nisan (2010) propose a $\frac{1}{2}$-approximation maximal in range (MIR) algorithm, which means maximizing the total social welfare on a set $\mathcal{X}'\subseteq\mathcal{X}$ yields at least $\frac{1}{2}$ of the maximal social welfare on the whole $\mathcal{X}$. We now explain the composition of the set of allocations $\mathcal{X}'$. We divide $h$ items into $n^2$ bundles of the same size $A=\lfloor\frac{h}{n^2}\rfloor$ and a possible distinct bundle with size $A'$ which contains all the remaining items. $\mathcal{X}'$ contains all allocations about these $O(n^2)$ bundles in the sense that all items in a bundle can only be simultaneously allocated. Finally the problem is converted to a knapsack problem and there exists an $\frac{1}{2}$-approximation algorithm that runs in $O(\text{poly}(n))$ time.

For the construction of $\Gamma$, we make some modifications to the original construction of Dudík et al. (2020) to get a binary-valued PTM. We first define $\mathcal{A}=\{mA+nA':m\in\{0,1,\dots,n^2\},n\in\{0,1\}\}$; note that $|\mathcal{A}|\leq 2n^2+2$. We denote $g_1,\dots,g_{|\mathcal{A}|}$ to be the elements of $\mathcal{A}$ in non-decreasing order. Then, we select $\Gamma$ to be a $|\mathcal{X}'|\times n|\mathcal{A}|$ matrix. For any allocation $a^{(k)}=[g_{\tau_1},\dots,g_{\tau_n}]$, $k\in[|\mathcal{X}'|]$, $j\in[n]$ and $\ell\in[|\mathcal{A}|]$, we define $\Gamma^{(k,i)}=\mathbb{I}[\tau_j>\ell]$ where $i=(j-1)|\mathcal{A}|+\ell$. Note that $\Gamma$ is 1-implementable because each column corresponds to a valid valuation function. In addition, we have the following result that bounds the regret with respect to the $1/2$-approximation of the best revenue in hindsight.

**Theorem 7.** *With the aforementioned $\Gamma$ in hand, we can combine Algorithm 2 with the $\frac{1}{2}$-approximate MIR algorithm in Dobzinski & Nisan (2010) and get the following regret bound:*

$$\mathbb{E}\left[\frac{1}{2}\left(\max_{a\in\mathcal{X}}\sum_{t=1}^{T}\sum_{i=1}^{n}b_t^{(i)}(a^{(i)})\right)-\sum_{t=1}^{T}\sum_{i=1}^{n}b_t^{(i)}(a_t^{(i)})\right]=O(n^3\sqrt{T-L_T^*}) \tag{52}$$

*Proof.* We first show that $\Gamma$ is $N$-approximable. Since $\Gamma$ is a binary matrix, by Lemma 3 it suffices to show that $\Gamma$ does not possess two identical rows. This can be verified by noticing that $\Gamma^{(k)}$ and $\Gamma^{(k')}$ are binary encodings of $a^{(k)}$ and $a^{(k')}$ by applying indicator functions.

Thus, the PTM $\Gamma$ is indeed $N$-approximable. By Corollary 1 we have

$$\mathbb{E}\left[\max_{a\in\mathcal{X}'}\left(\sum_{t=1}^{T}\sum_{i=1}^{n}b_t^{(i)}(a^{(i)})\right)-\sum_{t=1}^{T}\sum_{i=1}^{n}b_t^{(i)}(a_t^{(i)})\right]=O\left(\max\{\gamma,\ln K,\sqrt{N\ln K}\}\sqrt{T-L_T^*}\right)$$
$$=O\left(\max\{N,\ln K,\sqrt{N\ln K}\}\sqrt{T-L_T^*}\right)$$
$$=O\left(N\sqrt{T-L_T^*}\right)$$
$$=O\left(n|\mathcal{A}|\sqrt{T-L_T^*}\right)$$
$$=O(n^3\sqrt{T-L_T^*}). \tag{53}$$

Above, we use the fact that $\Gamma$ is binary-valued to get $N = \Omega(\log K)$, $N = n|\mathcal{A}|$ and $|\mathcal{A}| = O(n^2)$. Combining (53) with the fact that the best allocation in $\mathcal{X}'$ is a $\frac{1}{2}$-approximation to the best allocation in $\mathcal{X}$ yields the stated regret bound. $\qquad \square$

**Simultaneous second-price auctions** We now consider the *utility optimization* problem from the point of view of a *bidder* repeatedly participating in a simultaneous second-price auction (with different bidders each time). In this problem, $n$ bidders want to bid for $h$ items. Each bidder has a combinatorial valuation function $v$ to describe valuations for different bundles and submits a bid vector $b$ for all $h$ items. If he gets an allocation $q$, his payment profile is given $p$, where $p$ is the vector of the second highest bids. In particular, his utility is given by $u(b, p) = v(q) - p \cdot q$. Each round the bidder chooses a bidder vector and the adversary chooses the second largest bidder's vector. The goal is to find bidding vectors which compete with the best bidding vector in hindsight.

Following Dudík et al. (2020), we assume that both bids and the valuation function only take values in the discretized set $\left\{ 0, \frac{1}{m}, \ldots, \frac{m}{m} \right\}$. We also make the no-overbidding assumption that $v(q) \geq p \cdot q$ and denote the set of feasible bidding vectors to be $\mathcal{B}$. Let $\Gamma$ be a $\mathcal{B} \times hm$ matrix. For any $j \in [h]$, $\ell \in [m]$, denote $i = (j - 1)m + \ell$. For a bidding vector $b^{(k)} = [b^{(k,1)}, \ldots, b^{(k,h)}]$ and a vector of the second largest bids $p^{(i)} = \frac{\ell}{m} e_j + \sum_{j' \neq j} e_{j'}$, we set $\Gamma^{(k,i)} = \mathbb{I}\left[ b^{(k,j)} \geq \frac{\ell}{m} \right] = \frac{u(b^{(k)}, p^{(i)})}{v(e_j) - \frac{\ell}{m}}$. Note that this directly implies that $\Gamma$ is 1-implementable.

**Theorem 8.** *The aforementioned $\Gamma$ is $N$-approximable. Thus, running Algorithm 2 for the simultaneous second-price auction on the discretized set $\mathcal{B}$ yields*

$$\mathbb{E}\left[ \max_{b \in \mathcal{B}} \sum_{t=1}^{T} u(b, p_t) - \sum_{t=1}^{T} u(b_t, p_t) \right] = O(hm\sqrt{T - L_T^*}) \tag{54}$$

*Proof.* Notice that $\Gamma$ is binary and the rows of $\Gamma$ come from component-wise threshold functions of the bidding vector. Therefore, for two different bidding vectors the corresponding two rows in $\Gamma$ would also be different. Thus, we can apply Corollary 1 to get

$$\begin{aligned} \mathbb{E}\left[ \max_{b \in \mathcal{B}} \sum_{t=1}^{T} u(b, p_t) - \sum_{t=1}^{T} u(b_t, p_t) \right] &= O\left( \max\{\gamma, \ln K, \sqrt{N \ln K}\} \sqrt{T - L_T^*} \right) \\ &= O\left( \max\{N, \ln K, \sqrt{N \ln K}\} \sqrt{T - L_T^*} \right) \\ &= O\left( N\sqrt{T - L_T^*} \right) \\ &= O(hm\sqrt{T - L_T^*}). \end{aligned} \tag{55}$$

This completes the proof. $\qquad \square$

### B.3 Level auction

We consider the online level auction problem with single-item, $n$-bidders, $s$-level and $m$-discretization level. We give only a brief review of this problem setting, and we refer the reader to Dudík et al. (2020) for a complete description. In each round $t$ of this problem, firstly an auctioneer picks $s$ non-decreasing thresholds from a discretized set $\{\frac{1}{m}, \ldots, \frac{m}{m}\}$ for each bidder. Let $a_t = [a_t^{(1,1)}, \ldots, a_t^{(1,s)}, \ldots, a_t^{(n,1)}, \ldots, a_t^{(n,s)}]$ be the collection of the auctioneer's choices, where $a^{(i,j)}$ is $j$-th the threshold for the $i$-th bidder. Let $\mathcal{A} \in \{\frac{1}{m}, \ldots \frac{m}{m}\}^{ns}$ be all possible auctions. We further make the following assumption on $\mathcal{A}$, which corresponds to $\mathcal{S}_{s,m}$ as considered in (Dudík et al., 2020, Section 3.3).

**Assumption 1.** *Assume: (a) $\forall a, a' \in \mathcal{A}$, there exists at least one pair $(i, j) \in [n] \times [s]$, such that $a^{(i,j)} \neq a'^{(i,j)}$; and (b) $\forall a \in \mathcal{A}$, $\forall i \in [n]$, $a^{(i,1)} < \cdots < a^{(i,s)}$.*

After $a_t$ is chosen, the bidders reveal their valuations. We denote the collection of the bidders' valuations as $b_t = [b_t^{(1)}, \ldots, b_t^{(n)}] \in \mathcal{B} = [0, 1]^n$. As a consequence, the auctioneer obtains a reward $r(a_t, b_t)$, which is calculated based on the following rule.

**Definition 4** (The rule for level auction). *For each bidder $i \in [n]$, define the level index $\ell_t^{(i)}$ be the maximum $j$ such that $a_t^{(i,j)} \leq b_t^{(i)}$, with $\ell_t^{(i)} = 0$ when $a_t^{(i,1)} > b_t^{(i)}$. For each round $t$, all bidders whose level indexes are $0$ would be eliminated. If no bidder left, $r(a_t, b_t) = 0$. Otherwise, the bidder with the highest level index wins the item, and pays the price (i.e., $r(a_t, b_t)$) equal to the minimum bid that he could have submitted and still won the auction. On the same level, the tie-break rule is in favor of the bidder with the smallest bidding index.*

In this framework the set of "experts" is the set $\mathcal{A}$ of threshold configurations over all bidders. The regret of the auctioneer over $T$ iterations is the gap between the revenue generated by the online choice of threshold configurations and the revenue of the best set of thresholds in hindsight, i.e.

$$R_T := \mathbb{E}\left[\max_{a \in \mathcal{A}} \sum_{t=1}^{T} r(a, b_t) - \sum_{t=1}^{T} r(a_t, b_t)\right].$$

### B.3.1 Algorithm and regret

We first discuss how to construct an approximable (see Definition 2) and implementable $\Gamma$ with small $\gamma$. Directly constructing $\Gamma$ is rather difficult. Thus, we first consider an *augmented* auction problem with $n + 1$ bidders. Let $\mathcal{A}' \in \{\frac{1}{m}, \ldots, \frac{m}{m}\}^{(n+1)s}$ be the set of possible auctions, and $\mathcal{B}' \in [0, 1]^{n+1}$ be the set of bidder profiles. We construct $\mathcal{A}'$ as follows:

**Definition 5.** *(a) Distinct auctions (first $n$ bidders): $\forall a, a' \in \mathcal{A}'$, there exists at least one pair $(i, j) \in [n] \times [s]$, such that $a^{(i,j)} \neq a'^{(i,j)}$; (b) distinct thresholds (first $n$ bidders): $\forall a \in \mathcal{A}'$, $\forall i \in [n]$, $a^{(i,1)} < \cdots < a^{(i,s)}$; and (c) fixed thresholds for the $(n + 1)$-th bidder: $\forall a \in \mathcal{A}'$, $a^{(n+1,1)} = \frac{1}{m}$, $a^{(n+1,j)} = \frac{j-1}{m}$ for $j \in \{2, \ldots, s\}$.*

Comparing Assumption 1 and Definition 5, it can be seen that the elements in $\mathcal{A}'$ and $\mathcal{A}$ have an one-to-one correspondence: $\forall a \in \mathcal{A}$, there only exists one $a' \in \mathcal{A}'$, such that $\forall (i, j) \in [n] \times [s]$, $a^{(i,j)} = a'^{(i,j)}$, and vice versa.

In the augmented problem, at each round $t$, firstly the auctioneer chooses $a_t'$ from $\mathcal{A}'$. At the same time, we let the bidders reveal $b_t' = [b_t; 0]$, where $b_t$ is the bidder vector of the original problem, and $b_t'$ is a $(n + 1)$-dimensional vector. Then, the auctioneer obtain a reward $r'(a_t', b_t')$, where $r' : \mathcal{A}' \times \mathcal{B}' \mapsto [0, 1]$ follows the auction rule in Definition 4. For $a_t'$, denote the related auction in $\mathcal{A}$ as $a_t$, and we have the following lemma.

**Lemma 14.** *We have $r(a_t, b_t) = r'(a_t', b_t')$, and*

$$\mathbb{E}\left[\max_{a \in \mathcal{A}} \sum_{t=1}^{T} r(a, b_t) - \sum_{t=1}^{T} r(a_t, b_t)\right] = \mathbb{E}\left[\max_{a' \in \mathcal{A}'} \sum_{t=1}^{T} r'(a', b_t') - \sum_{t=1}^{T} r'(a_t', b_t')\right].$$

*Proof.* Since the last element of $b_t'$ is $0$, so the $(n + 1)$-th bidder will always be at the $0$-th level when computing $r'(a_t', b_t')$, it will not change the reward at round $t$. We note that, it does not mean the augmentation is not useful: designing a PTM for $r'(a', b')$ is much easier than for the original problem. $\square$

This lemma reveals a duality between the two problems: A low-regret and oracle-efficient algorithm for the augmented problem directly induces a low-regret and oracle-efficient algorithm for the original problem by replacing $a' \in \mathcal{A}'$ with its corresponding auction $a_t$ in $\mathcal{A}$. To help understanding, we illustrate the relationship between the original problem and the augmented problem in Figure 1.

Next, we propose to construct an approximable $\Gamma$ matrix for the augmented problem by using $ns(m - s + 1)$ bid profiles in $\mathcal{B}'$. For $i \in \{1, \ldots, n\}$, $j \in \{1, \ldots, s\}$, and $k \in \{1, \ldots, m - s + 1\}$, let a bidder vector $b'^{(i,j,k)} \in \mathcal{B}'$ be

$$b'^{(i,j,k)} = \frac{k + j - 1}{m} e_i + \frac{j - 1}{m} e_{n+1}, \tag{56}$$

where $e_i$ is a $(n + 1)$-dimension unit vector whose $i$-th element is $1$. Note that the construction here is to ensure that only the $i$-th and the $(n + 1)$-th bidders are likely to win, which greatly simplifies our construction of the PTM. The following lemma illustrates how to construct the PTM and the corresponding vector $s$ which satisfy the $\gamma$-approximability condition.

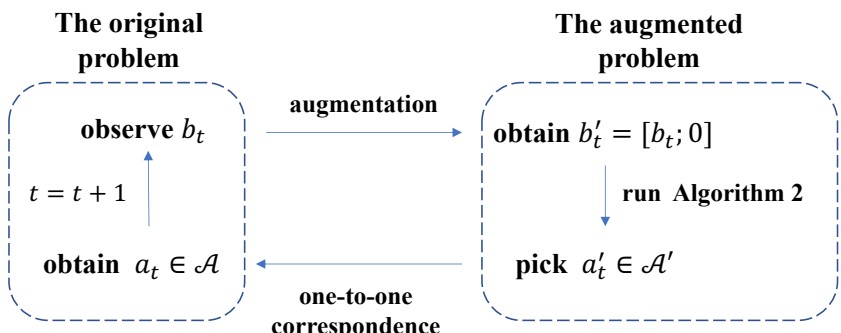

Figure 1: The relationship between the original problem and our proposed augmented problem.

**Lemma 15.** *Let $\mathcal{V} = \{b'^{(i,j,k)}\}_{i,j,k}$ be the set containing all $b'$ defined in (56). Let $\Gamma^{\mathcal{V}} \in [0,1]^{K \times ns(m-s+1)}$ be the matrix implemented from $\mathcal{V}$ by assigning each $r'(\cdot, b'^{(i,j,k)})$ to the columns of $\Gamma^{\mathcal{V}}$ one-by-one. Then $\Gamma^{\mathcal{V}}$ is $nsm$-approximable.*

*Proof.* We first prove $r'(a', b'^{(i,j,k)}) = a'^{(i,j)}\mathbb{I}[a'^{(i,j)} \leq \frac{k+j-1}{m}] + \frac{j-1}{m}\mathbb{I}[a'^{(i,j)} > \frac{k+j-1}{m}]$, then argue that it leads to $nsm$-approximability.

For $j = 1$, $b'^{(i,1,k)} = \frac{k}{m}e_i$. According to Definition 4, when $a'^{(i,1)} > \frac{k}{m}$, all bidders are at the 0-th level, so no bidder wins the item, and $r(a', b'^{(i,j,k)}) = 0$. Otherwise, bidder $i$ wins the item, and pays $a'^{(i,1)}$. Thus, $r'(a', b'^{(i,1,k)}) = a'^{(i,1)}\mathbb{I}[a'^{(i,1)} \leq \frac{k}{m}]$. For $j > 1$, $b'^{(i,j,k)} = \frac{k+j-1}{m}e_i + \frac{j-1}{m}e_{n+1}$. Based on the third part of Definition 5, bidder $n+1$ is at the $j$-th level. If $a'^{(i,j)} > \frac{k-j+1}{m}$, then bidder $n+1$ wins the item, and pays $r'(a', b'^{(i,j,k)}) = \frac{j-1}{m}$. Otherwise, bidder $i$ wins the item, and pays $r'(a', b'^{(i,j,k)}) = a'^{(i,j)}$. In sum, we have $r'(a', b'^{(i,j,k)}) = a'^{(i,j)}\mathbb{I}[a'^{(i,j)} \leq \frac{k+j-1}{m}] + \frac{j-1}{m}\mathbb{I}[a'^{(i,j)} > \frac{k+j-1}{m}]$ holds for any $j \in [s]$.

Next, we prove the approximability based on Lemma 13. WLOG, consider one auction $a' \in \mathcal{A}'$, and let $\Gamma^{\mathcal{V},a'}$ be the row related to $a'$, which is a $ns(m-s+1)$-dimensional vector based on Lemma 15. Our goal is to show that, there exists a vector $s$, such that

$$\forall \widehat{a}' \in \mathcal{A}', \widehat{a}' \neq a', \left\langle \Gamma^{\mathcal{V},a'} - \Gamma^{\mathcal{V},a}, s \right\rangle \geq 1.$$

Denote $s^{(i,j,k)}$ as the element of $s$ which is related to $b'^{(i,j,k)}$ (see Lemma 15), and we discuss how to set $s^{(i,j,k)}$ as follows.

First, based on the second part of Definition 5, we have $\frac{j}{m} \leq a'^{(i,j)} \leq \frac{m-s+j}{m}$. Thus, combining (56), we know $\forall i \in [n], j \in [s], \exists k' \in [m-s+1]$, such that $b'^{(i,j,k')} = a'^{(i,j)}e_i + \frac{j-1}{m}e_{n+1}$, which is in turn equivalent to $\frac{k'+j-1}{m} = a'^{(i,j)}$. In the column corresponding to $b'^{(i,j,k')}$, $\forall \widehat{a}' \in \mathcal{A}', \widehat{a}'^{(i,j)} \neq a'^{(i,j)}$, we have $r'(a', b'^{(i,j,k')}) - r'(\widehat{a}', b'^{(i,j,k')}) \geq \frac{1}{m}$. Intuitively, it means that, in this column, only auctions whose $j$-th threshold for the $i$-th bidder equals to $a'^{(i,j)}$ yield the highest revenue, and these auctions outperform other auctions by least $\frac{1}{m}$. Choosing the corresponding $s^{(i,j,k')}$ as $m$, and setting $s^{(i,j,k)} = 0$ for $k \neq k'$, makes $\sum_{k=1}^{m-s+1} r'(a', b'^{(i,j,k)})s^{(i,j,k)} - r'(\widehat{a}', b'^{(i,j,k)})s^{(i,j,k)} \geq 1$. Since we need to ensure that the loss gap between $a'$ and any other auctions by at least 1, we need to set $s^{(i,j,k')} = m$ for any $i \in [n], j \in [s]$. It is obvious that $\|s\|_1 = nsm$ and $\Gamma^{\mathcal{V}}$ is $nsm$-approximable according to Lemma 13. $\qquad\square$

To illustrate the construction of $\Gamma^{\mathcal{V}}$, we provide an example for the case where $m = 5$ and $s = 3$ and inspect encodings with respect to a single bidder in Table 1.

Combining Lemma 15, Corollary 1, we get the following results.

Table 1: Illustration of $\Gamma^{\mathcal{V}}$ when $m = 5$ and $s = 3$. Each $a'^{(i,j)}$ is encoded by $m - s + 1 = 3$ columns. Here $g = \frac{j-1}{m} = \frac{2-1}{5} = \frac{1}{5}$, and $h = \frac{j-1}{m} = \frac{3-1}{5} = \frac{2}{5}$. Consider the auctions whose $a'^{(i,2)} = \frac{3}{5} = \frac{k'+j-1}{m} = \frac{k'+1}{5}$. Then, at the second column that is related to $a'^{(i,2)}$ (corresponds to $k' = 2$), only such auctions can yield the highest revenue $\frac{3}{5}$, and the revenue is at least $\frac{1}{5}$ higher than auctions with $a'^{(i,2)} \neq \frac{3}{5}$.

| | Auction a' | | | | Coding | | | | | | | | |
|---|---|---|---|---|---|---|---|---|---|---|---|---|---|
| | $(a'^{(i,1)}, a'^{(i,2)}, a'^{(i,3)})$ | | | | $a'^{(i,1)}$ | | | $a'^{(i,2)}$ | | | $a'^{(i,3)}$ | | |
| ... | | ... | ... | | | | | | | | | | ... |
| ... | $(1/5, 2/5, 3/5)$ | ... | ... | 1/5 | 1/5 | 1/5 | 2/5 | 2/5 | 2/5 | 3/5 | 3/5 | 3/5 | ... |
| ... | $(1/5, 2/5, 4/5)$ | ... | ... | 1/5 | 1/5 | 1/5 | 2/5 | 2/5 | 2/5 | h | 4/5 | 4/5 | ... |
| ... | $(1/5, 2/5, 5/5)$ | ... | ... | 1/5 | 1/5 | 1/5 | 2/5 | 2/5 | 2/5 | h | h | 5/5 | ... |
| ... | $(1/5, 3/5, 4/5)$ | ... | ... | 1/5 | 1/5 | 1/5 | g | 3/5 | 3/5 | h | 4/5 | 4/5 | ... |
| ... | $(1/5, 3/5, 5/5)$ | ... | ... | 1/5 | 1/5 | 1/5 | g | 3/5 | 3/5 | h | h | 5/5 | ... |
| ... | $(1/5, 4/5, 5/5)$ | ... | ... | 1/5 | 1/5 | 1/5 | g | g | 4/5 | h | h | 5/5 | ... |
| ... | $(2/5, 3/5, 4/5)$ | ... | ... | 0 | 2/5 | 2/5 | g | 3/5 | 3/5 | h | 4/5 | 4/5 | ... |
| ... | $(2/5, 3/5, 5/5)$ | ... | ... | 0 | 2/5 | 2/5 | g | 3/5 | 3/5 | h | h | 5/5 | ... |
| ... | $(2/5, 4/5, 5/5)$ | ... | ... | 0 | 2/5 | 2/5 | g | g | 4/5 | h | h | 5/5 | ... |
| ... | $(3/5, 4/5, 5/5)$ | ... | ... | 0 | 0 | 3/5 | g | g | 4/5 | h | h | 5/5 | ... |
| ... | ... | ... | ... | ... | ... | ... | ... | ... | ... | ... | ... | ... | ... |

**Corollary 2.** *Consider running Algorithm 2 with $\Gamma^{\mathcal{V}}$ on the augmented problem. Let $\{a'_t\}_t \in \mathcal{A}'^T$ be the output of the algorithm, and $\{a_t\}_t \in \mathcal{A}^T$ be the corresponding auctions in the original problem. Then Algorithm 2 is oracle-efficient, and*

$$\mathbb{E}\left[\sum_{t=1}^{T} r(a^*, b_t) - \sum_{t=1}^{T} r(a_t, b_t)\right] = O\left(\max\left\{nsm, ns\ln m, \sqrt{nms \cdot ns\ln m}\right\}\sqrt{T - L_T^*}\right)$$

$$= O\left(nsm\sqrt{T - L_T^*}\right).$$

(57)

## B.4 Proof of Lemma 4

In this section, we prove Lemma 4, which yields oracle-efficient online learning with a very small output space. Recall that $\mathcal{X} = \{x^{(1)}, \ldots, x^{(K)}\}$ and we denote the adversary's output space as $\mathcal{Y} = \{y^{(1)}, \ldots, y^{(d)}\}$. We construct $\Gamma$ as $\forall k \in [K], j \in [d]$,

$$\Gamma^{(k,j)} = f(x^{(k)}, y^{(j)}).$$

It is straightforward to see that in this way $\Gamma$ is implementable with complexity 1. On the other hand, for each $y_t$, we can find $j_t \in [d]$, such that $y^{(j_t)} = y_t$. Thus, we can meet 1-approximability by choosing $s = e_{j_t}$ for action $k \in [K]$ in round $t$, where $e_t$ is a unit vector whose $j_t$-th dimension is 1 and all other elements are 0. This completes the proof. □

## B.5 Proof of Lemma 5

In this section, we prove Lemma 5, which yields oracle-efficient online learning for the transductive online classification problem. For this problem, we create a PTM $\Gamma$ with $|\mathcal{W}|$ columns, which is configured as $\Gamma^{(k,j)} = f(x^{(k)}, (w^{(j)}, 1)), \forall k \in [K], j \in [|\mathcal{W}|]$.

It is clear that $\Gamma$ is implementable with complexity 1. Next, we prove that $\Gamma$ approximable. Let $w_t$ be the feature vector observed in round $t$. Then there exists $j_t \in [|\mathcal{W}|]$ such that $w^{(j_t)} = w_t$. If $y_t = 1$, then the equation of Definition 2 holds by setting $s = e_{j_t}$. If $y_t = 0$, then the equation can be met by picking $s = -e_{j_t}$. This completes the proof of the lemma. It is worth noting that this choice of $\Gamma$ need not be $\delta$-admissible for any $\delta > 0$.

## B.6 Negative Implementability

When the oracle only accepts non-negative weights for minimizing the loss (or non-positive weights for maximizing the reward), Algorithms 1 and 2 cannot make use of the oracle directly, since the

---

**Algorithm 4** Oracle-based GFTPL with negative exponential distribution

---

1: **Input:** Data set $S_j$, $j \in [N]$, that implement a matrix $\Gamma \in [0,1]^{K \times N}$, $\eta_1 = \min\{\frac{1}{\gamma}, 1\}$.
2: Draw IID vector $\widehat{\alpha} \sim \text{Exp}(1)^N$, and let $\alpha = [\alpha^{(1)}, \dots, \alpha^{(N)}] = -\widehat{\alpha}$
3: **for** $t = 1, \dots, T$ **do**
4:    Choose $x_t \leftarrow \underset{k \in [K]}{\operatorname{argmin}} \sum_{j=1}^{t-1} f(x^{(k)}, y_j) + \sum_{i=1}^{N} \frac{\alpha^{(i)}}{\eta_t} \left[ \sum_{(w,y) \in \mathcal{S}_i} w \cdot r(x^{(k)}, y) \right]$
5:    Observe $y_t$
6:    Compute $\widehat{L}_t^* = \min_{k \in [K]} \sum_{j=1}^{t} f(x^{(k)}, y_j)$ by using the oracle, set $\eta_{t+1} \leftarrow \min\left\{ \frac{1}{\gamma}, \frac{1}{\sqrt{\widehat{L}_t^*+1}} \right\}$
7: **end for**

---

noise $\alpha$ can be negative for Algorithm 1, and positive for Algorithm 2. To handle this issue, we consider two solutions: (a) constructing *negative-implementable* PTMs (first defined by Dudík et al. (2020)); (b) replacing the distribution of $\alpha$ from the Laplace distribution with the (negative) exponential distribution. The former solution can be used in VCG with bidder-specific reserves, envy-free item pricing, problems with small $\mathcal{Y}$ and transductive online classification, while the latter is suitable for the level auction problem and multi-unit online welfare maximization.

### B.6.1 Negative implementable PTM

To deal with negative weights, Dudík et al. (2020) introduce the concept of negative implementability:

**Definition 6.** *A matrix $\Gamma$ is negatively implementable with complexity $M$ if for each $j \in [N]$ there exist a (non-negatively) weighted dataset $S_j^-$, with $|S_j^-| \leq M$, such that $\forall i, i' \in [K]$,*

$$-(\Gamma^{(i,j)} - \Gamma^{(i',j)}) = \sum_{(w,y) \in \mathbb{R}_+ \times \mathcal{Y}} w(f(x^{(i)}, y) - f(x^{(i')}, y)).$$

Similar to Theorem 5.11 of Dudík et al. (2020), we have the following theorem.

**Theorem 9.** *Suppose the oracle can only accept non-negative weights (for minimizing the loss). If $\Gamma$ is implementable and negative implementable with complexity $M$, then Algorithm 1 can achieve oracle-efficiency with per-round complexity $O(T + NM)$.*

For VCG with bidder-specific reserves and envy-free item pricing, (Dudík et al., 2020) show that there exist *binary* and admissible PTMs that are implementable and negative implementable. Then, based on Lemma 3, this kind of PTMs directly leads to approximable, implementable and negative implementable PTMs, so the oracle-efficiency and the small-loss bound can be achieved for these settings according to the theorem above.

Moreover, we can also find approximable, implementable and negative-implementable PTMs in the other mentioned applications of a) problems with a small output space $\mathcal{Y}$ and b) transductive online classification. For application a) recall that we constructed $\Gamma$ as $\Gamma^{(i,j)} = f(x^{(i)}, y^{(j)})$. Then, it is straightforward to verify that this matrix can be negatively implemented by setting $\Gamma^{(i,j)} = 1 - f(x^{(i)}, y^{(j)})$. For application b), recall that we set $\Gamma^{(k,j)} = f(x^{(k)}, (w^{(j)}, 1))$, $\forall k \in [K], j \in [|\mathcal{W}|]$. This PTM can be negatively implemented by simply setting $\Gamma^{(k,j)} = f(x^{(k)}, (w^{(j)}, 0))$, $\forall k \in [K], j \in [|\mathcal{W}|]$, which flips all elements of the binary matrix (for 0 to 1 and 1 to 0).

### B.6.2 Negative exponential distribution

For the other auctions problems that we consider in this paper (i.e. multi-unit mechanisms and level auctions), the PTMs that we constructed are not negative implementable. However, we show that for these cases, Algorithm 2 with a negative exponential distribution is good enough to achieve the small-loss bound. This adjusted algorithm is summarized in Algorithm 4. Note that Algorithm 4 can directly use the oracle because $\alpha$ is non-positive.

For Algorithm 4, we introduce the following theorem. Our key observation is that, in these settings, the approximable vector (i.e., the vector $s$ in Definition 2) is *always element-wise non-negative*, which makes the proof go through when using the negative exponential distribution.

**Theorem 10.** *Let $f(x,y) = 1 - r(x,y)$. Assume $\Gamma$ is $\gamma$-approximable w.r.t. $f(x,y)$ and implementable with function $r(x,y)$. Moreover, suppose $\forall y \in [\mathcal{Y}]$, $k \in [K]$, the approximable vector $s$ is element-wise non-negative. Then Algorithm 4 is oracle-efficient and achieves the following regret bound:*

$$R_T = \mathbb{E}\left[L_T^* - \sum_{t=1}^{T} r(x_t, y_t)\right] = O\left(\max\{\gamma, \ln K, N\}\sqrt{T - L_T^*}\right).$$

*Proof.* The proof is similar to that of Theorem 1, and here we only provide the sketch of the proof. For the relation between $\mathbb{P}[x_t = x^{(i)}]$ and $\mathbb{P}[x_t' = x^{(i)}]$, similar to (6), we still have $\mathbb{P}[x_t = x^{(i)}] \leq \exp(\eta_t \gamma)\mathbb{P}[x_t' = x^{(i)}]$. The difference lies in (7): $\beta$ (corresponds to $\alpha$ in Algorithm 4) therein has support on the non-positive orthant, since $s$ is non-negative, $\frac{p(\beta)}{p(\beta-s)}$ is always well-defined. Thus, the negative exponential distribution is enough to make the proof go through. Next, for the upper bound of TERM 1, let $\widehat{\alpha} = -\alpha$ be the exponential distribution. Similar to the proof of Lemma 7, we have

$$2\gamma \sum_{t=1}^{T} \eta_t \mathbb{E}[f(x_t', y_t)]$$

$$\overset{(1)}{\leq} 2\gamma \sum_{t=1}^{T} \eta_t \mathbb{E}\left[\left(\sum_{j=1}^{t} f(x_{t+1}, y_j) + \left\langle \Gamma^{(x_{t+1})}, \alpha_{t+1}\right\rangle\right) - \left(\sum_{j=1}^{t-1} f(x_t, y_j) + \left\langle \Gamma^{(x_t)}, \alpha_t\right\rangle\right)\right]$$

$$+ 2\gamma \sum_{t=1}^{T} \eta_t \left(\frac{1}{\eta_t} - \frac{1}{\eta_{t+1}}\right)\mathbb{E}\left[\Gamma^{(x_{t+1})}\alpha\right]$$

$$\overset{(2)}{\leq} 2\gamma \sum_{t=1}^{T} \eta_t \mathbb{E}\left[\left(\sum_{j=1}^{t} f(x_{t+1}, y_j) + \left\langle \Gamma^{(x_{t+1})}, \alpha_{t+1}\right\rangle\right) - \left(\sum_{j=1}^{t-1} f(x_t, y_j) + \left\langle \Gamma^{(x_t)}, \alpha_t\right\rangle\right)\right]$$

$$+ 2\gamma \sum_{t=1}^{T} \eta_t \left(\frac{1}{\eta_{t+1}} - \frac{1}{\eta_t}\right)\mathbb{E}\left[\max_{i\in[K]} \Gamma^i \widehat{\alpha}\right]$$

$$\overset{(3)}{\leq} 2\gamma\eta_T \cdot \mathbb{E}\left[\sum_{j=1}^{T} f(x^*, y_j) + \left\langle \Gamma^{(x^*)}, \alpha_{T+1}\right\rangle\right] + 2\gamma \sum_{t=1}^{T} \eta_t \left(\frac{1}{\eta_{t+1}} - \frac{1}{\eta_t}\right)\mathbb{E}\left[\max_{i\in[K]} \Gamma^i \widehat{\alpha}\right]$$

$$+ 2\gamma \sum_{t=1}^{T-1}(\eta_{t-1} - \eta_t) \cdot \mathbb{E}\left[\sum_{j=1}^{t-1} f(x^*, y_j) + \left\langle \Gamma^{(x^*)}, \alpha_t\right\rangle\right] + 2\gamma\eta_1 \cdot \mathbb{E}\left[\max_{i\in[K]} \Gamma^{(i)} \widehat{\alpha}\right]$$

$$\overset{(4)}{\leq} 2\gamma\eta_T L_T^* + 2\gamma \sum_{t=1}^{T-1}(\eta_{t-1} - \eta_t)L_{t-1}^* + 2\gamma \sum_{t=1}^{T-1}(\eta_{t-1} - \eta_t)\mathbb{E}\left[\max_{i\in[K]} \Gamma^i \widehat{\alpha}\right]$$

$$+ 2\gamma \sum_{t=1}^{T} \eta_t \left(\frac{1}{\eta_{t+1}} - \frac{1}{\eta_t}\right)\mathbb{E}\left[\max_{i\in[K]} \Gamma^i \widehat{\alpha}\right] + 4\gamma\eta_1 \mathbb{E}\left[\max_{i\in[K]} \Gamma^i \widehat{\alpha}\right],$$

where inequality (1) follows from (12), inequality (2) is because $\widehat{\alpha} = -\alpha$ and $\widehat{\alpha}$ is non-negative, inequaliry (3) is due the the optimality of $x_t$, and the final inequality (4) is based on the non-negativity of $\widehat{\alpha}$. Note that there are some extra terms since the distribution is no longer zero-mean. To proceed, the second term can be upper bounded by (13). For the third term, similar to (15), we have $\forall \lambda \leq 1/2$,

$$\mathbb{E}\left[\max_{i\in[K]} \Gamma^i \widehat{\alpha}\right] \leq \frac{1}{\lambda}\ln\left(K\left(\frac{1}{1-\lambda}\right)^N\right) = \frac{\ln K}{\lambda} + \frac{N}{\lambda}\ln\left(\frac{1}{1-\lambda}\right) \leq 4\max\{\ln K, N\}, \quad (58)$$

where in the first inequality we make use of the moment generating function of the exponential distribution. Thus, we have

$$2\gamma \sum_{t=1}^{T-1} (\eta_{t-1} - \eta_t)\mathbb{E}\left[\max_{i\in[K]} \Gamma^i \widehat{\alpha}\right] \leq 8\max\{\ln K, N\},$$

and

$$2\gamma \sum_{t=1}^{T} \eta_t \left(\frac{1}{\eta_{t+1}} - \frac{1}{\eta_t}\right)\mathbb{E}\left[\max_{i\in[K]} \Gamma^i \widehat{\alpha}\right] \leq \frac{8\max\{\ln K, N\}}{\eta_{T+1}}.$$

The proof can be finished by applying (58) and similar techniques as in the proof of Lemma 10 to bound TERM 2. $\qquad\square$

Finally, we note that, as shown in the proof of Lemma 15, the approximable vector for the level auction problem is element-wise non-negative (0-1 vector), so Theorem 10 can be directly applied. In the following, we show that this conclusion can also be applied to the online welfare maximization for multi-unit items.

**Lemma 16.** *For multi-unit online welfare maximization, there exists an approximable vector $s$ with non-negative entries $\forall a^{(k)} \in \mathcal{X}', k \in [K]$.*

*Proof.* By Lemma 13, it suffices to prove that for any $k \in [K]$ there exists non-negative $s$ such that $\langle \Gamma^{(k)} - \Gamma^{(j)}, s\rangle \geq 1$. For multi-unit online welfare maximization (as illustrated in Appendix B.2), all $h$ items need to be allocated. There do not exist two rows $\Gamma^{(k)}$ and $\Gamma^{(k')}$ such that $\Gamma^{(k)} \preceq \Gamma^{(k')}$ because the corresponding allocations $a^{(k)}$ and $a^{(k')}$ also preserve this partial order relation, which means for allocation $a^{(k)}$ there are unassigned items. We can simply take $s = \Gamma^{(k)}$. Based on the aforementioned observation, there exists at least one index $\ell \in [N]$ such that $\Gamma^{(k,\ell)} = 1$ and $\Gamma^{(j,\ell)} = 0$, and thus $\langle \Gamma^{(k)} - \Gamma^{(j)}, s\rangle \geq 1$.

$\qquad\square$

## C   Proof of Theorem 3

In this section we prove Theorem 3, which is our oracle-efficient "best-of-both-worlds" bound, assuming that the adversary is oblivious. Then, based on the definitions, we know $\widehat{U}_T^{\mathsf{FTL}}$ and $\widehat{U}_T^{\mathsf{GFTPL}}$ only depend on the adversary (i.e., the past losses), and is independent of the randomness of the algorithm. This also applies to $\mathcal{I}_T^{\mathsf{FTL}}$ and $\mathcal{I}_T^{\mathsf{GFTPL}}$.

The regret can be decomposed into two parts:

$$
\begin{aligned}
R_T^{\mathsf{OFF}} &= \mathbb{E}\left[\sum_{t=1}^{T} f(x_t, y_t) - \sum_{t=1}^{T} f(x^*, y_t)\right] \\
&\leq \mathbb{E}\left[\sum_{t\in\mathcal{I}_T^{\mathsf{FTL}}} f(x_t, y_t) - f(x_T^{\mathsf{FTL},*})\right] + \mathbb{E}\left[\sum_{t\in\mathcal{I}_T^{\mathsf{GFTPL}}} f(x_t, y_t) - f(x_T^{\mathsf{GFTPL},*})\right] \qquad (59) \\
&\leq \widehat{U}_T^{\mathsf{FTL}} + \widehat{U}_T^{\mathsf{GFTPL}},
\end{aligned}
$$

where

$$x_T^{\mathsf{FTL},*} = \operatorname*{argmin}_{i\in[K]} \sum_{t\in\mathcal{I}_T^{\mathsf{FTL}}} f(x^{(i)}, y_t),$$

and

$$x_T^{\mathsf{GFTPL},*} = \operatorname*{argmin}_{i\in[K]} \sum_{t\in\mathcal{I}_T^{\mathsf{GFTPL}}} f(x^{(i)}, y_t).$$

In round $T$, there are four possible cases:

- Case 1: $\mathsf{Alg}_T = \mathsf{FTL}$, and $\mathsf{Alg}_{T+1} = \mathsf{FTL}$.

  Since after round $T$, the algorithm does not switch, we have $\widehat{U}_T^{\mathsf{FTL}} \leq \alpha \widehat{U}_T^{\mathsf{GFTPL}}$ based on lines 4-8. On the other hand, let $t'$ be the last round where the algorithm performs GFTPL, that is, $\mathsf{Alg}_{t'+1} = \mathsf{FTL}$. Then, in round $t' - 1$, if we do the switch $\mathsf{FTL} \to \mathsf{GFTPL}$, then

  $$\alpha \widehat{U}_{t'-1}^{\mathsf{GFTPL}} \leq \widehat{U}_{t'-1}^{\mathsf{FTL}}.$$

  Moreover, note that $\widehat{U}_t^{\mathsf{FTL}}$ and $\widehat{U}_t^{\mathsf{GFTPL}}$ are non-decreasing, and also $\widehat{U}_{t'-1}^{\mathsf{GFTPL}} \geq \widehat{U}_{t'}^{\mathsf{GFTPL}} - \tau$. Combining with the fact that $\widehat{U}_{t'}^{\mathsf{GFTPL}} = \widehat{U}_T^{\mathsf{GFTPL}}$ (since we do not feed losses to GFTPL from round $t'$ to $T$), we have

  $$\alpha(\widehat{U}_T^{\mathsf{GFTPL}} - \tau) = \alpha(\widehat{U}_{t'}^{\mathsf{GFTPL}} - \tau) \leq \alpha \widehat{U}_{t'-1}^{\mathsf{GFTPL}} \leq \widehat{U}_{t'-1}^{\mathsf{FTL}} \leq \widehat{U}_T^{\mathsf{FTL}},$$

  so

  $$\widehat{U}_T^{\mathsf{GFTPL}} \leq \frac{1}{\alpha} \widehat{U}_T^{\mathsf{FTL}} + \tau.$$

  If in round $t' - 1$ we use GFTPL and do not switch, then we have

  $$\frac{1}{\beta} \widehat{U}_{t'-1}^{\mathsf{GFTPL}} \leq \widehat{U}_{t'-1}^{\mathsf{FTL}},$$

  thus

  $$\frac{1}{\beta}(\widehat{U}_T^{\mathsf{GFTPL}} - \tau) = \frac{1}{\beta}(\widehat{U}_{t'}^{\mathsf{GFTPL}} - \tau) \leq \frac{1}{\beta} \widehat{U}_{t'-1}^{\mathsf{GFTPL}} \leq \widehat{U}_{t'-1}^{\mathsf{FTL}} \leq \widehat{U}_T^{\mathsf{FTL}},$$

  which implies that

  $$\widehat{U}_T^{\mathsf{GFTPL}} \leq \beta \widehat{U}_T^{\mathsf{FTL}} + \tau.$$

- Case 2: $\mathsf{Alg}_T = \mathsf{FTL}$, and $\mathsf{Alg}_{T+1} = \mathsf{GFTPL}$.

  Since after round $T$, we have $\mathsf{FTL} \to \mathsf{GFTPL}$, we get $\widehat{U}_T^{\mathsf{FTL}} > \alpha \widehat{U}_T^{\mathsf{GFTPL}}$ based on lines 4-8. On the other hand, We know that $\mathsf{Alg}_T = \mathsf{FTL}$, so after round $T - 1$, there are 2 possibilities: 1) the algorithm remains to be FTL. For this case, we have

  $$\widehat{U}_T^{\mathsf{FTL}} - 1 \leq \widehat{U}_{T-1}^{\mathsf{FTL}} \leq \alpha \widehat{U}_{T-1}^{\mathsf{GFTPL}} \leq \alpha \widehat{U}_T^{\mathsf{GFTPL}},$$

  where we use the fact that the mixability gap $\delta_t \leq 1$. It yields

  $$\widehat{U}_T^{\mathsf{FTL}} \leq \alpha \widehat{U}_T^{\mathsf{GFTPL}} + 1.$$

  2) The algorithm switches from $\mathsf{GFTPL} \to \mathsf{FTL}$. For this case, we have

  $$\widehat{U}_T^{\mathsf{FTL}} - 1 \leq \widehat{U}_{T-1}^{\mathsf{FTL}} \leq \frac{1}{\beta} \widehat{U}_{T-1}^{\mathsf{GFTPL}} \leq \frac{1}{\beta} \widehat{U}_T^{\mathsf{GFTPL}},$$

  so

  $$\widehat{U}_T^{\mathsf{FTL}} \leq \frac{1}{\beta} \widehat{U}_T^{\mathsf{GFTPL}} + 1.$$

- Case 3: $\mathsf{Alg}_T = \mathsf{GFTPL}$, and $\mathsf{Alg}_{T+1} = \mathsf{FTL}$.

  Since after round $T$, we switch from $\mathsf{GFTPL} \to \mathsf{FTL}$, we have $\widehat{U}_T^{\mathsf{FTL}} \leq \frac{1}{\beta} \widehat{U}_T^{\mathsf{GFTPL}}$. On the other hand, in round $T-1$, there are 2 cases: 1) After round $T - 1$, we switch the algorithm: $\mathsf{FTL} \to \mathsf{GFTPL}$. Thus,

  $$\widehat{U}_T^{\mathsf{FTL}} \geq \widehat{U}_{T-1}^{\mathsf{FTL}} \geq \alpha \widehat{U}_{T-1}^{\mathsf{GFTPL}} \geq \alpha(\widehat{U}_T^{\mathsf{GFTPL}} - \tau),$$

  implying that

  $$\widehat{U}_T^{\mathsf{GFTPL}} \leq \frac{1}{\alpha} \widehat{U}_T^{\mathsf{FTL}} + \tau.$$

  2) After round $T - 1$, the algorithm does not switch: $\mathsf{GFTPL} \to \mathsf{GFTPL}$. Thus,

  $$\frac{1}{\beta}(\widehat{U}_T^{\mathsf{GFTPL}} - \tau) \leq \frac{1}{\beta} \widehat{U}_{T-1}^{\mathsf{GFTPL}} \leq \widehat{U}_{T-1}^{\mathsf{FTL}} \leq \widehat{U}_T^{\mathsf{FTL}},$$

  so $\widehat{U}_T^{\mathsf{GFTPL}} \leq \beta \widehat{U}_T^{\mathsf{FTL}} + \tau$.

- Case 4: $\mathsf{Alg}_T = \mathsf{GFTPL}$, $\mathsf{Alg}_{T+1} = \mathsf{GFTPL}$.
  For this case, after round $T$, we have

$$\widehat{U}_T^{\mathsf{GFTPL}} \le \beta \widehat{U}_T^{\mathsf{FTL}}.$$

On the other hand, let $t'$ be the last round of the algorithm that plays $\mathsf{FTL}$. So in round $t' - 1$, there are 2 possible cases: 1) After $t' - 1$, we switch from $\mathsf{GFTPL} \to \mathsf{FTL}$. In this case, we have:

$$\beta(\widehat{U}_T^{\mathsf{FTL}} - 1) = \beta(\widehat{U}_{t'}^{\mathsf{FTL}} - 1) \le \beta \widehat{U}_{t'-1}^{\mathsf{FTL}} \le \widehat{U}_{t'-1}^{\mathsf{GFTPL}} \le \widehat{U}_T^{\mathsf{GFTPL}},$$

so

$$\widehat{U}_T^{\mathsf{FTL}} \le \frac{1}{\beta} \widehat{U}_T^{\mathsf{GFTPL}} + 1.$$

2) After round $t' - 1$, we still play $\mathsf{FTL}$. Then, we have

$$\widehat{U}_T^{\mathsf{FTL}} - 1 = \widehat{U}_{t'}^{\mathsf{FTL}} - 1 \le \widehat{U}_{t'-1}^{\mathsf{FTL}} \le \alpha \widehat{U}_{t'-1}^{\mathsf{GFTPL}} \le \alpha \widehat{U}_T^{\mathsf{GFTPL}},$$

so

$$\widehat{U}_T^{\mathsf{FTL}} \le \alpha \widehat{U}_T^{\mathsf{GFTPL}} + 1.$$

Combining all of the pieces, we always have

$$\widehat{U}_T^{\mathsf{FTL}} \le (\alpha + \frac{1}{\beta}) \widehat{U}_T^{\mathsf{GFTPL}} + 1,$$

and

$$\widehat{U}_T^{\mathsf{GFTPL}} \le (\frac{1}{\alpha} + \beta) \widehat{U}_T^{\mathsf{FTL}} + \tau.$$

Finally, note that based on the definition, it is straightforward to get $\widehat{U}_T^{\mathsf{GFTPL}} \le U_T^{\mathsf{GFTPL}}$ and $\widehat{U}_T^{\mathsf{FTL}} \le U_T^{\mathsf{FTL}}$. Therefore, we have

$$R_T^{\mathsf{OFF}} \le \min\left\{ \left(1 + \alpha + \frac{1}{\beta}\right) U_T^{\mathsf{GFTPL}} + 1, \left(1 + \frac{1}{\alpha} + \beta\right) U_T^{\mathsf{FTL}} + \tau \right\}.$$

Setting $\alpha = \beta = 1$ yields the required theorem. $\qquad \square$