# OpenReview forum: "Adaptive Oracle-Efficient Online Learning"
_NeurIPS.cc/2022/Conference — NeurIPS 2022 Accept_

### Official Review · Reviewer_z8qi · 2022-07-09

**Rating:** 7
**Confidence:** 3
**Soundness:** 3 good
**Presentation:** 3 good
**Contribution:** 3 good

**Summary:**

The paper's main contribution is an FTPL-style algorithm that achieves a small-loss bound in the full-information setting, leading to the first Oracle efficient algorithm to achieve such a guarantee. The key insight leading to this result is the observation that ensuring that $P[x_t\neq x_{t+1}]$ is small (something that can be achieved for the outputs $x_1, x_2, \dots$ of existing FTPL algorithms) is not enough to achieve a small loss bound. The authors identify a new condition on the perturbation process called approximability that ensures that the ratio $P[x_t \neq x^i]/P[x_{t+1}\neq x^i]$ is bounded, which they show is sufficient for achieving a small loss bound. The authors identify many applications where this condition on the noise process is satisfied.

**Questions:**

Some questions are included in the limitations section.

**Limitations:**

See suggestions in the strengths and weaknesses section. Here are some typos:
- Line 94. condition -> conditions
- Line 139. The use of "On the other hand" does not make sense here.
- Line 146. You cite Van Erven at. al 2014 when you mention FTPL. I do not think they were the ones to invent it. Perhaps more appropriate to cite Kalai and Vempala 2005 or something earlier.
- Line 204. It is easy to that for -> It is easy to see that for.
- Line 225. time-variant -> time-varying
- Line 244. has a a -> has a
- In the display within Cor 1, there is an \epsilon that is not scoped.
- Line 266. "Noticing the fact that N= \Omega (ln K)". Where are we supposed to notice this from?
- Line 316. Your notion of mixability gap is different from that of Rooij et al 2014.

**Strengths And Weaknesses:**

The new approximability condition on the noise process of FTPL that the paper presents seems natural and leads to new, non-trivial results in Oracle efficient Online Learning. This is a strong contribution in my opinion. Weaknesses if any would be in the writing; in particular, the order in which some concepts are presented in the paper can be improved in my opinion.
- First of all, it would be helpful to spell out explicitly early on that the point of using a PTM is to reduce computation as much as possible by only requiring a noise vector of dimension much smaller than the number of experts (you do explain this, but my point is just that it would be better if this point is crystal clear early on---the explanation on line 66 is not clear enough for someone who hasn't read the rest of the paper).
- On line 200, you say that a PTM is not compatible with the Oracle, which is rather vague. It is not until you see the definition of an implementable PTM that things make sense. I would suggest mentioning something about implementable earlier in the paper.
- On line 177, you mention "However, motivated by adaptive FTPL in the inefficient setting". Here by inefficient, you mean requiring a noise vector of the dimension the same as the number of experts. Spell this out.

---

> ### Author Response · Authors · 2022-08-02
> **Response to Reviewer z8qi**
>
> Thank you very much for the positive assessment of our work as well as all of the detailed and constructive comments regarding the presentation! We appreciate your catching the mentioned typos and will fix these immediately. We address a couple of your specific questions/comments below:
>
> >“Line 266. "Noticing the fact that N= \Omega (ln K)". Where are we supposed to notice this from?”
>
> Note that when the PTM Gamma is binary and 1-admissible, every two rows of Gamma differ by at least one element. This means that Gamma must, at the very least, include \Omega (ln K) columns to encode each row. We will explain this in more detail in the revision.
>
> >“Your notion of the mixability gap is different from that of Rooij et al 2014.”
>
> Thank you for the careful comparison. Just to clarify, we use the specific expression for the mixability gap in (the second paragraph, Page 1286 of Rooij et al, 2014 [1]). We will be more specific about the citation in Lemma 6 of the paper to avoid this confusion.
>
> [1]https://homepages.cwi.nl/~pdg/ftp/flipflop.pdf

---

### Official Review · Reviewer_c3ru · 2022-07-11

**Rating:** 5
**Confidence:** 3
**Soundness:** 3 good
**Presentation:** 3 good
**Contribution:** 2 fair

**Summary:**

This paper considers online learning problems given offline optimization oracles and provide an algorithm that requires only a small number of oracle calls.
The main proposed algorithm has a cumulative-loss dependent regret bound, which implies improved performances for small-loss environments.
The algorithm is extended to establish best-of-both-worlds bounds, which means that it works at least as well as the follow-the-leader algorithm.

**Questions:**

- Is my understanding above correct? For example, if the application of Flipfrop is non-trivial and in need of special consideration, I would appreciate it if you could highlight it.

- Is this the first study dealing with the FTPL approach with time-dependent adaptive parameters $\eta_t$? If so, that would be strong support for the novelty of this paper.


**Limitations:**


The limitations are adequately addressed.

**Strengths And Weaknesses:**

Strengths:

- The paper provides the first oracle-efficient algorithm with a small-loss regret bound for online learning problem.
- Assumptions and results are clearly stated

Weaknesses:

- Novelty in algorithms and analysis techniques is somewhat limited. Algorithms 1 and 2 are minor modification of algorithms by (Dud{\'\i}k et al., 2020). Algorithm 3 appears to be a direct application of (De Rooij et al., 2014).

Comments:

This study provides the first oracle-efficient algorithm with a small-loss regret bound for online learning problem.
The proposed algorithm is based on that by (Dud{\'\i}k et al., 2020).
The assumption of admissibility and implementablity also follows that by (Dud{\'\i}k et al., 2020).
The novelty lies in the adaptive updating of the parameter $\eta_t$ corresponding to the learning rate and in the introduction of the concept of approximability for the analysis.
A best-of-both-worlds bound is also achieved by a direct application of Flipflop approach by (De Rooij et al., 2014).

While there has been steady progress as a result, the impression is that it is merely a combination of existing results.
Unless this impression is dispelled in future discussions, I cannot strongly support the acceptance.

---

> ### Author Response · Authors · 2022-08-02
> **Response to Reviewer c3ru**
>
> Thank you for the review and comments! We appreciate the opportunity to clarify some possible misunderstandings below.
>
> > “Algorithms 1 and 2 are minor modification of algorithms by (Dud{'\i}k et al., 2020). Algorithm 3 appears to be a direct application of (De Rooij et al., 2014).”
>
> We address Algorithm 3 later in this response, and address Algorithms 1 and 2 here. It is worth noting that Algorithms 1 and 2 are not minor modifications of G-FTPL (Dudik et al, 2020), and our analysis also deviates significantly from the purely worst-case analysis presented in (Dudik et al, 2020). First, G-FTPL uses a bounded-support uniform distribution on the perturbation noise. It turns out not to be possible to prove an adaptive small-loss bound on the original G-FTPL algorithm (or, indeed, even the original FTPL algorithm) with bounded-support noise; this is for related reasons to the fundamental limitations of the admissibility condition that we show in Lemma 1 of our paper. We instead use DP-style distributions (e.g. Laplace noise) for the perturbation noise, which play well with the new approximability condition that we introduce. We also imbue the algorithm with a time-varying adaptive step size, which has not been done before in the oracle-efficient online learning paradigm.
>
> Our analysis is also very different from (Dudik et al, 2020). Lemma 1 shows fundamental limitations of the admissibility condition introduced by (Dudik et al, 2020) in showing adaptive small-loss bounds. Therefore, naively modifying the analysis of G-FTPL (Dud{'\i}k et al., 2020) does not work. We successfully address this problem by introducing a new condition for PTM, i.e., approximability. Compared with admissibility, our new condition ensures stronger stability, which makes it works well with DP-style distributions and leads to the stronger small-loss bound. Apart from figuring out the right condition for the small-loss bound, another challenging question is the design of approximable PTMs in real-world applications. In section 4, we utilize several novel techniques for constructing approximable PTMs in different problems and for proving the approximability.
>
>
>
> >“The assumption of admissibility and implementability also follows that by (Dud{'\i}k et al., 2020).”
>
> This may be a misunderstanding; the admissibility assumption is not used in the present work, we only present it for comparison. Our counterexamples in Lemma 1 show that admissibility is insufficient for proving small-loss bounds. In lieu of admissibility we introduce the new condition of approximability that is sufficient for small-loss bounds; this can also allow for PTMs that are, in fact, not admissible. As a result, our implementable PTMs need to be approximable rather than admissible, leading to significantly different PTM constructions for many real-world applications.
>
> >“Is my understanding above correct? For example, if the application of Flipfrop is non-trivial and in need of special consideration, I would appreciate it if you could highlight it.”
>
> Algorithm 3 does resemble Flipflop at a high level, i.e., by switching the algorithms based on a direct performance comparison. However, there are notable differences in the details of the algorithm as well as analysis, listed below:
> 1) The Flipflop algorithm of (De Rooij et al., 2014) is specifically designed for combining AdaHedge and Hedge, in which the meta-algorithm relies on comparing the “mixability gap” of two algorithms. It has two major limitations: a) the computation of the mixability gap involves all experts and is thus not oracle-efficient; b) it cannot be directly used to combine other methods, because the mixability gap only makes sense for AdaHedge and Hedge.
> 2) Our meta-algorithm instead directly compares the regret upper bounds of the base algorithms, and it can combine any two algorithms with any-time regret bounds. As a result, the algorithm is oracle-efficient whenever a) the base algorithms are oracle-efficient, and b) the regret bounds of the base algorithms can be computed efficiently. We apply our meta-algorithm combining G-FTPL and FTL as base algorithms.
> 3) To our knowledge, we are the first to show that the idea of Flipflop can be extended to combine any two algorithms with any-time regret bounds. We believe our analysis, while relatively simple, is of independent interest and can be applied to other problems in the realm of adaptive online learning.
>
>
> >“Is this the first study dealing with the FTPL approach with time-dependent adaptive parameters eta_t?”
>
> Yes, our work is the first to imbue G-FTPL with a time-variant step size (while preserving its oracle-efficiency).

---

> > ### Comment · Reviewer_c3ru · 2022-08-08
> > **Thank you for your clarification**
> >
> > Thank you for your clarification.
> > I now have a better understanding of the contributions of this study and have corrected some of my misunderstandings.
> > I am generally satisfied with the responses and am now leaning towards increasing my score, but will update it after the reviewer discussion period.

---

### Official Review · Reviewer_2kab · 2022-07-12

**Rating:** 6
**Confidence:** 3
**Soundness:** 3 good
**Presentation:** 3 good
**Contribution:** 3 good

**Summary:**

The paper studies the online learning problem with general (possibly non-convex) losses but a finite number K  of possible actions (i.e., the experts setting). Prior works obtain regret bounds that have the optimal dependence on T. A different line of work, such as that for convex losses, has shown that one can obtain adaptive, problem-dependent,  regret bounds that are much better than the worst-case bounds when the losses are small or iid. The current paper asks whether such adaptive bounds can be obtained in the expert settings using algorithms that are oracle efficient in the sense that the running time has sublinear dependency on the number K of possible actions, with a logarithmic dependency or better being particularly suited for exponentially-sized action spaces.  The paper adapts the algorithm and analysis of Dudik et al., and obtain the first adaptive oracle-efficient algorithms.

**Questions:**

None

**Strengths And Weaknesses:**

Originality: In order to obtain an oracle-efficient algorithm that adapts to the small losses, the paper adapts the approach of Dudik et al. A key bottleneck is designing a suitable perturbation matrix. The paper shows that the types of perturbation matrices considered by Dudik et al. are not suitable for the adaptive setting. To overcome this, the paper introduces an alternative type of perturbation matrices that are related to but incomparable to those considered by Dudik et al. The paper also shows that several practically relevant applications admit suitable perturbation matrices and thus the framework can be applied to all of these settings. By building on the work of Rooij et al., the paper also gives an algorithm for iid losses. Although the paper builds extensively on prior works, the new components needed to extend those works to the adaptive setting seem to be sufficiently novel.

Quality: The paper seems to be theoretically sound. Both the theoretical and empirical claims seem to be well supported.

Clarity: The main body of the paper is sufficiently clear and well written.

Significance: The paper gives the first adaptive algorithms that are oracle-efficient in the experts settings. The resulting framework has several important applications. Both the theoretical results and the practical applications seems to be good contributions to the online learning literature.

---

> ### Author Response · Authors · 2022-08-02
> **Response to Reviewer 2kab**
>
> Thank you for the constructive review and supportive comments about our work!

---

### Official Review · Reviewer_dDpe · 2022-07-13

**Rating:** 7
**Confidence:** 3
**Soundness:** 3 good
**Presentation:** 3 good
**Contribution:** 2 fair

**Summary:**


Summary: One of the primary contribution of this work was to provide a new framework for designing follow-the-perturbed leader algorithms that are oracle-efficient and adapt well to the small-loss environment, under "approximability". Authors also extend their results to an IID data setting and establish a “best-of-both-worlds” bound in the oracle-efficient setting.


**Questions:**

The results for best-of-both setting is not at all clear. Entire Sec. 5 is not elaborated well, what are the two different problem environments (adversarial and stochastic ??), what is the regret benchmarked against in both cases, and how does Thm 3 claims that Alg3 achieves simultaneously achieves best of both both both setups?


**Limitations:**

See Qs above

**Strengths And Weaknesses:**


- Strong theoretical results: Oracle efficient algorithms with regret guarantees
- Applications

Weakness: Scopes of the work appears to be limited since often in the real world problem, coming up with an efficient oracle might not be possible. Papers addresses few applications but for relatively simple/finite/structured settings.
Also the best-of both results (Sec 5) are not well defined/explained.

---

> ### Author Response · Authors · 2022-08-02
> **Response to Reviewer dDpe**
>
> Thank you very much for the review and positive comments about our theoretical results! We address your specific questions and concerns below.
>
> >“Scopes of the work appears to be limited since often in the real world problem, coming up with an efficient oracle might not be possible. Papers address few applications but for relatively simple/finite/structured settings.”
>
> Efficient optimization sub-routines exist across a broad range of real-world applications, including a) empirical risk minimization for supervised machine learning, b) data-driven market design (e.g. Nisan & Ronen, 2007) and c) dynamic programming (Bertsekas, 2019). These sub-routines are directly applicable on stochastic data; the paradigm of oracle-efficient online learning assumes access to these sub-routines as optimization oracles and broadens their applicability to non-stochastic data. The focus of this paper is to achieve adaptive guarantees for oracle-efficient online learning, and we present examples of applications across all of these domains. We also note that the oracle-efficient online learning paradigm could be applied to approximation algorithms (see, e.g. Niazadeh et al, 2021), further broadening its applicability.
>
> ---
>
> >“The results for best-of-both setting is not at all clear. Entire Sec. 5 is not elaborated well, what are the two different problem environments (adversarial and stochastic ??), what is the regret benchmarked against in both cases, and how does Thm 3 claims that Alg3 achieves simultaneously achieves best setups?”
>
>  Thank you for the feedback on Section 5; we will be sure to improve the writing in this section as per your suggestions. To address your specific questions: The regret is measured with respect to the best fixed expert in hindsight in both cases. The FTL algorithm achieves much better regret for stochastic data; in particular, when the loss of each expert is iid (with different means for each expert), the expected regret of FTL becomes a constant. On the other hand, the G-FTPL algorithm achieves better regret under adversarial data. Theorem 3 shows that the regret of Algorithm 3 is the minimum of the regret bounds of G-FTPL and FTL. Thus, it automatically achieves the much better constant regret bound of FTL under iid data without knowing the presence of stochasticity in data beforehand. We will add to the discussion under Theorem 3 to make these points clearer.

---

### Author Response · Authors · 2022-08-08
**Any questions?**

Dear reviewers,

We would like to thank you again for your helpful and constructive comments. Please let us know if you have any other questions, and we are more than happy to clarify. Thank you!

Best,\
Authors

---

### Meta-Review · Area_Chair_Q8RN · 2022-08-24

**Recommendation:** Accept
**Confidence:** Certain

**Metareview:**

The paper received reviews from experts in online learning, who all support acceptance following some clarifications provided by the authors.  From my own look into the paper, I also firmly support acceptance: the paper makes a clear, solid and elegant contribution to a long line of research in online learning, and it is also very well written.  I do however strongly encourage the authors to pay close attention to the suggestions in the reviews as to how to improve their presentation for the final version.

**Award:**

No

---

### Decision · Program_Chairs · 2022-09-14

Accept